# Measurements and models of electric fields in the *in vivo* human brain during transcranial electric stimulation

Yu Huang[1†], Anli A Liu[2*†], Belen Lafon[1], Daniel Friedman[2], Michael Dayan[3], Xiuyuan Wang[2], Marom Bikson[1], Werner K Doyle[2], Orrin Devinsky[2], Lucas C Parra[1*]

[1]Department of Biomedical Engineering, City College of the City University of New York, New York, United States; [2]Comprehensive Epilepsy Center, New York University School of Medicine, New York, United States; [3]Department of Neurology, Mayo Clinic, Rochester, United States

**Abstract** Transcranial electric stimulation aims to stimulate the brain by applying weak electrical currents at the scalp. However, the magnitude and spatial distribution of electric fields in the human brain are unknown. We measured electric potentials intracranially in ten epilepsy patients and estimated electric fields across the entire brain by leveraging calibrated current-flow models. When stimulating at 2 mA, cortical electric fields reach 0.8 V/m, the lower limit of effectiveness in animal studies. When individual whole-head anatomy is considered, the predicted electric field magnitudes correlate with the recorded values in cortical ($r = 0.86$) and depth ($r = 0.88$) electrodes. Accurate models require adjustment of tissue conductivity values reported in the literature, but accuracy is not improved when incorporating white matter anisotropy or different skull compartments. This is the first study to validate and calibrate current-flow models with *in vivo* intracranial recordings in humans, providing a solid foundation to target stimulation and interpret clinical trials.

**\*For correspondence:** anli.liu@
nyumc.org (AAL); parra@ccny.
cuny.edu (LCP)

[†]These authors contributed equally to this work

## Introduction

Transcranial electric stimulation (TES) delivers weak electric currents to the scalp with the goal of modulating endogenous brain activity (*Ruffini et al., 2013*). Stimulation can be applied as constant unidirectional current (tDCS) or biphasic alternating current (tACS). Weak constant direct currents (tDCS, $\leq 2$ mA) can elicit prolonged shifts in cortical excitability (*Nitsche and Paulus, 2000, 2001*; *Nitsche et al., 2003*). More recently weak alternating currents (tACS) have also been used in an effort to entrain or modulate brain activity (*Herrmann et al., 2013*; *Reato et al., 2013*; *Ali et al., 2013*; *Alekseichuk et al., 2016*; *Lustenberger et al., 2016*). Because of their simplicity, flexibility and safety profile, these techniques have been investigated in over 70 neuropsychiatric conditions, including major depression (*Bikson et al., 2008*), epilepsy (*Fregni et al., 2006d*; *Auvichayapat et al., 2013*), tinnitus (*Frank et al., 2012*), Parkinson's disease (*Fregni et al., 2006b*), pain control (*Fregni et al., 2006a, 2006c, 2007*), and stroke rehabilitation (*Schlaug et al., 2008*; *Baker et al., 2010*) among others. In healthy subjects tDCS may benefit declarative memory (*Marshall et al., 2004*), working memory (*Fregni et al., 2005*), motor learning (*Reis and Fritsch, 2011*), verbal fluency (*Pereira et al., 2013*), and planning ability (*Dockery et al., 2009*).

Many clinical trials place stimulation electrodes on the scalp based on the assumption that the 'active' electrode will stimulate the underlying brain region with a uniform polarity and irrespective of the placement of the 'return' electrode. Computational models of current flow have called this

simplistic approach into question (*Bikson et al., 2010*). These models capture anatomical details obtained with magnetic resonance imaging (MRI) to predict electric fields inside the brain. Models have shown that the polarity of stimulation is necessarily mixed due to cortical folding. Furthermore, nearby electrodes result in maximal electric fields between the two stimulation electrodes (*Datta et al., 2009*), and not underneath the electrodes as commonly assumed. Detailed computational models also suggest that more focused stimulation (1 cm radius) of the cortex may be achieved by using multiple smaller electrodes (*Datta et al., 2008*; *Dmochowski et al., 2011*; *Edwards et al., 2013*). Finally, these models predict that cerebrospinal fluid (CSF) in the inter-hemispheric and ventricular spaces could guide current to deep structures of the brain (*Datta et al., 2011*; *Senço et al., 2015*).

Unfortunately, despite increasing sophistication in these computational models (*Wagner et al., 2004*; *Datta et al., 2009*; *Sadleir et al., 2010*; *Parazzini et al., 2011*; *Datta et al., 2012*; *Minhas et al., 2012*; *Datta et al., 2010*; *Wagner et al., 2007*; *Datta et al., 2011*; *Dmochowski et al., 2013*; *Luu et al., 2016*), none of these model predictions have been directly validated to-date. Early validation efforts for simple spherical models used *ex vivo* recordings (*Rush and Driscoll, 1968*), surface recordings (*Burger and Milaan, 1943*) or *in vivo* recordings in simian (*Hayes, 1950*). Modern models have compared predictions to human scalp surface recordings (*Bangera et al., 2010*; *Datta et al., 2013*), but detailed models have not been validated with intracranial recordings *in vivo*. Furthermore, the models depend heavily on tissue conductivity values, which have only been measured *ex vivo* in other animal species (*Gabriel et al., 1996*; *Baumann et al., 1997*; *Oostendorp et al., 2000*; *Peters et al., 2001*; *Hoekema et al., 2003*). These measures differ from *in vivo* measurements in humans obtained over half a century ago (*Burger and Milaan, 1943*; *Freygang and Landau, 1955*).

As a result, there are substantial knowledge gaps in determining the electric fields realistically achieved in the brain. This is of particular concern given that the present estimates (of at most 1 V/m) are at the lower end of what is required to affect neuronal activity in animal experiments (*Reato et al., 2013*). There is also no certainty as to whether the spatial distribution of electric field intensities across the brain is sufficiently well predicted to warrant model-based targeting. Here we aim to address these questions with *in vivo* intracranial recordings in humans by directly measuring field intensities produced by TES at the cortical surface and deeper brain areas. We report results on ten (10) patients undergoing invasive monitoring for epilepsy surgery, with subdural grids, strips, and depth electrodes. In total we recorded from 1380 intracranial electrodes. These recordings are then compared to various detailed computational models, including differential conductivity between skull spongiosa and compacta, and white matter anisotropy. In doing so, we obtain calibrated models that conclusively answer outstanding questions about stimulation magnitudes, spatial distribution, and modeling choices. For instance, we find that maximal electric field magnitudes are approximately 0.8 V/m when using the generally accepted maximum of 2 mA scalp stimulation. These low field intensities are below the levels typically used in animal studies providing a challenge to future studies on the mechanisms of action of TES. To facilitate validation of new modeling approaches we have made all relevant measurement data publicly available (at 10.6080/K0XW4GQ1).

## Results

### Intracranial voltage recordings: linearity, frequency dependence, stability

Transcranial sinusoidal alternating current was applied to ten patients through two electrodes placed medially over the frontal and occipital pole (patients P03–P011 and P014, *Figure 1*). For one patient (P014) electrodes were placed at three additional locations (*Figure 2A*). We first confirmed that measured voltages increased linearly with current intensity (*Figure 3A*; up to saturation levels of the clinical amplifiers). A modest drop in magnitude was observed with increasing frequency (25% at 100 Hz; *Figure 3B*), which we ascribe to a non-uniform gain across frequencies for the measurement equipment and electrodes as confirmed with recording of voltages in saline. Thus, we observed uniform gain for the tissue indicating purely resistive medium in this frequency range. Repeated measurements with fixed frequency but minutes of separation indicate the level of stability of recordings

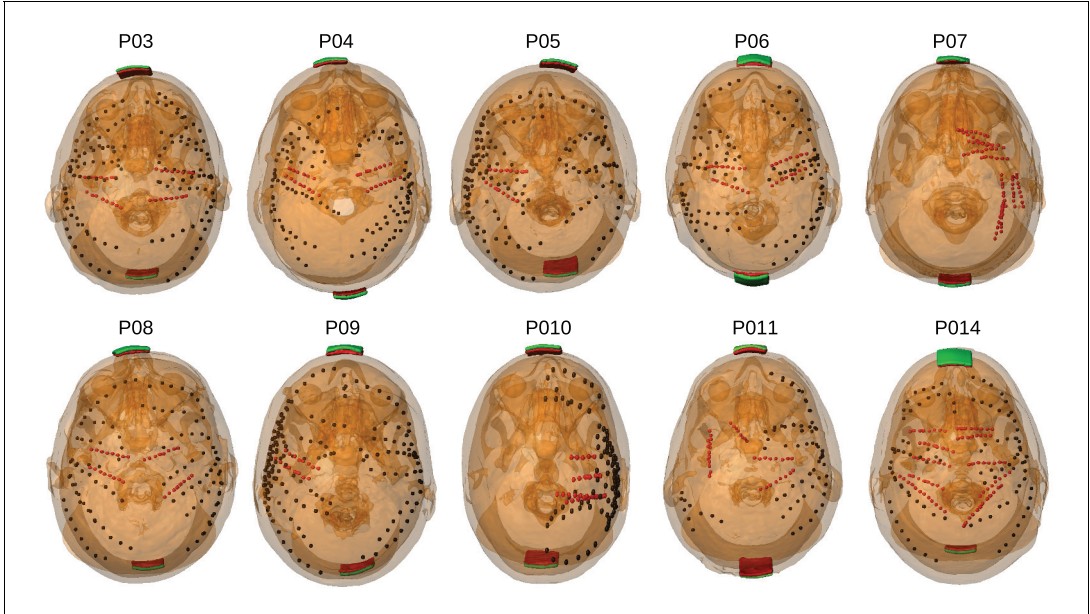

**Figure 1.** Location of the invasive recording electrodes and transcranial electrical stimulation electrodes in the 10 patients tested. Electrodes measuring from the cortical surface (64-contact grids, 8-contact strips) are indicated as black dots and depth electrodes (between 6–8 contacts each) as red dots. Square stimulation electrodes on scalp surface (2 cm), are shown in green with contact gel in red. Individual anatomy derived from the T1-weighted MRI is transparent to visualize electrode locations.

over time (*Figure 3C*). The observed drift in voltage magnitudes over time could have a number of sources (*Noury et al., 2016*), most likely patient movement. Regardless, the variability observed here sets the lower limit in precision one can expect to ±0.11 mV for 1 mA stimulation (standard deviation pooled across electrodes and subjects; *Figure 3C*, values for P03). Subsequent recordings were performed at 1 Hz and with stimulation current of 0.25 mA to 1 mA, limited by the dynamic range of the clinical amplifier and patient sensation.

## Measurements and predictions of electric field

Placement of the invasive recording electrodes was determined by clinical considerations. These electrodes do not necessarily capture electric field magnitudes at the locations with maximal stimulation intensities, nor will they be oriented to best capture the electric field in the orientation of the field vectors. To estimate electric field magnitude and orientation across the brain we built for each subject a realistic anatomical model of conductivity in the head (*Figure 4*) and calibrated the model by matching measurements with the predicted electric fields (see 'Conductivity optimization'). Recorded voltages and their distribution across the cortical surface are shown in *Figure 5A/B* for the recorded data and the calibrated, realistic head models. Animated 3D rendering of these images (A-C) are provided as supplementary material for all the subjects (rich media, see *Figure 5—source data 1*). For this subject (P03) the measured voltages are tightly correlated across locations with the predicted electric potential values. The same is true for all subjects resulting in high Pearson correlation coefficients, $r = 0.94 \pm 0.04$ (Here and in the following all summary statistics indicate mean and standard deviation across subjects). To determine the electric fields at each electrode we calculated the difference of the recorded voltages between neighboring electrodes divided by inter-electrode distance (see 'Voltage and projected electric field measurements'). This provides a measure of the local electric field along the direction of a given electrode pair in units of V/m, which we refer to as the 'projected electric field'. The measured and predicted values of this projected electric field are shown for one subject in *Figure 5E*. As expected, the correlation of predicted and measured electric fields is lower than for the raw potentials (here $r = 0.86$, $p = 10^{-14}$, $N = 45$), as the calculated field is the difference of two close-by measurements, each with some inherent noise. Similar results are obtained for all ten subjects ($r = 0.81 \pm 0.09$), suggesting that the spatial distribution of electric field

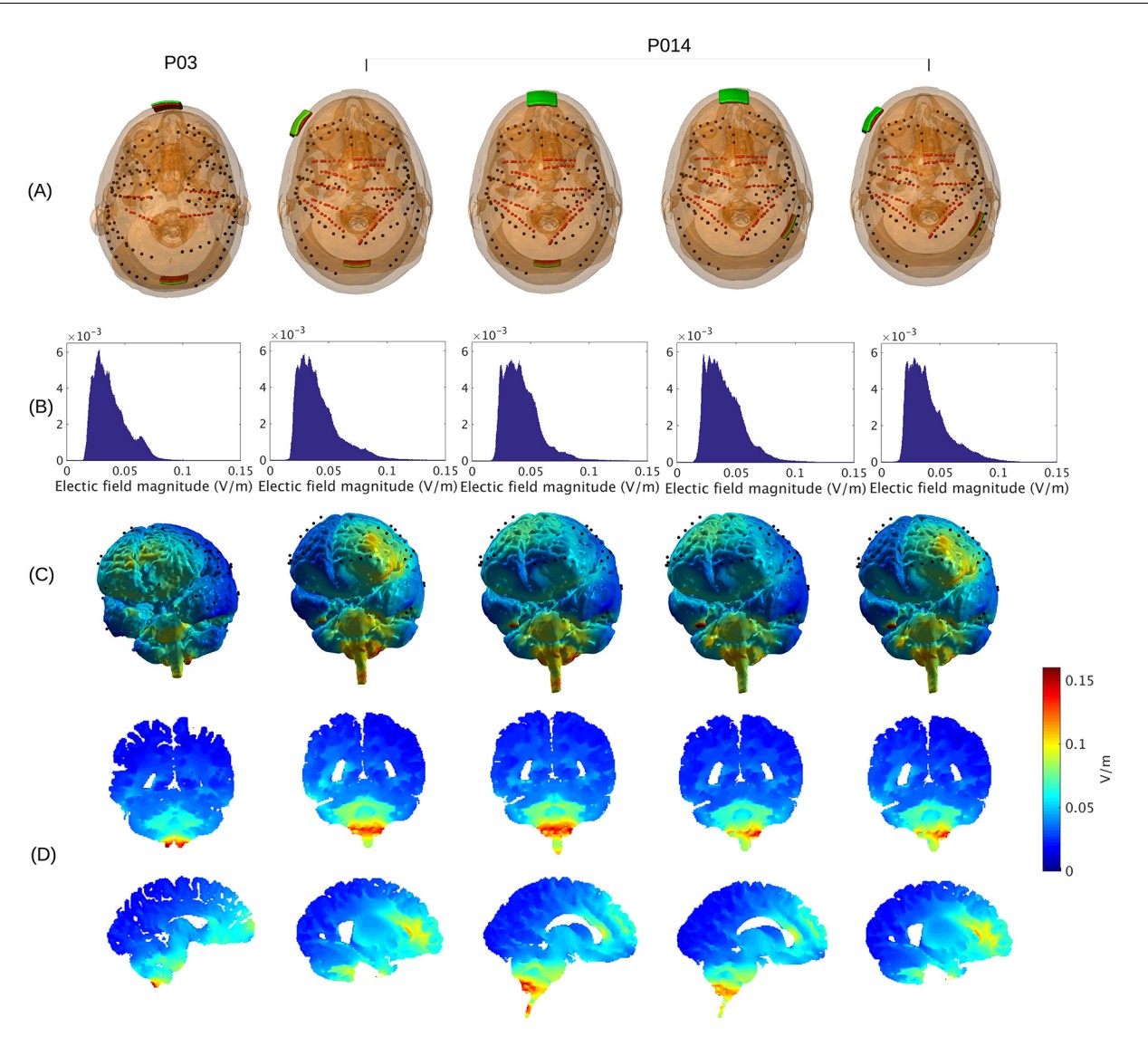

**Figure 2.** Prediction of electric field with calibrated models for various electrode montages at 1 mA stimulation intensity. (B) Histogram of electric field magnitude for the montage used on Subject P03 (same as in Figure 5) and Subject P014. (C) Corresponding spatial distributions on cortical surface. (D) Cross-section plots showing predicted electric field intensity in mid-brain areas with hot spots underneath stimulation electrodes and adjacent to highly conducting ventricles.

magnitudes is well predicted by the models. When collapsing all recordings across subjects (*Figure 5F/G*) we find correlation between measured and predicted field projections of $r = 0.86$ ($p = 10^{-118}$, $N = 405$) and $r = 0.88$ ($p = 10^{-54}$, $N = 164$) for cortical and depth electrodes respectively. Note also that the measured field projections in depth electrodes are nearly as strong as in cortical electrodes (*Figure 5F* vs. *Figure 5G*; standard deviation of measured field projections across electrodes are 0.059 V/m and 0.065 V/m, respectively).

## Estimated stimulation magnitude across the brain

Now that we have established the accuracy of the model predictions, we can estimate the electric field magnitudes produced throughout the brain. We ask, in particular, what are the maximum intensities achieved at the cortical surface and in deep brain regions? The electrode configuration used in the experiments gives a range of values within and across individuals (*Figure 2B*). All values reported

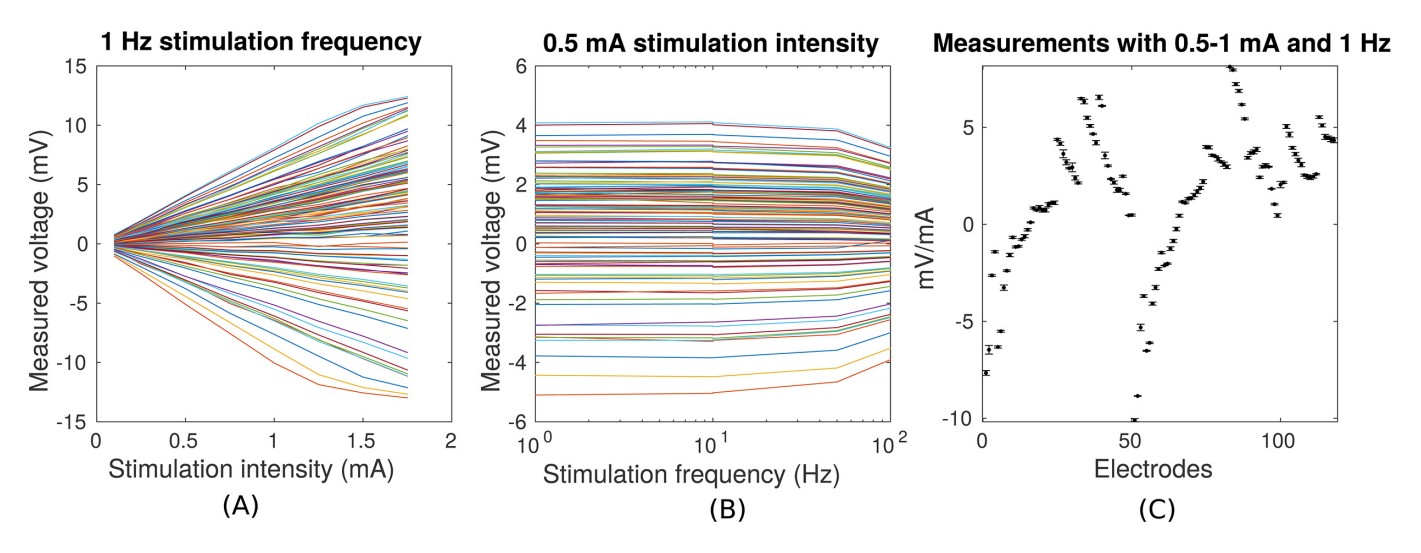

**Figure 3.** Voltage recordings across multiple intracranial locations for sinusoidal transcranial alternating current stimulation for the first subject tested (P03). Magnitude and sign are estimated by fitting a sinusoid to the voltage fluctuations at each electrode location. (**A**) Voltage recordings at multiple intracranial recording locations are linear with stimulation intensity up to 1 mA in this subject (each curve represents a different electrode). At higher intensities some channels saturate due to a limited dynamic range of the clinical recording equipment, which is demonstrated by the plateauing of measured voltage at intensities above 1.5 mA. (**B**) Intensities are constant with frequency in the range of 1–10 Hz. The drop-off at higher frequencies is due to the recording equipment. (**C**) Averaged measurements across three stimulation sessions (separated by approximately 1 min each) demonstrate stability of electric field measurements across sessions. (Here stimulation was 1 Hz and between 0.5–1 mA in stimulation current. The voltage values are calibrated to correspond to 1 mA stimulation). Error bars at each electrode indicate the variability across different stimulation blocks.

here and in *Figure 2* are scaled to 1 mA of stimulation. As expected for distant electrodes (*Dmochowski et al., 2011*; *2012*), strongest electric fields are achieved under the stimulating electrodes (e.g. under the occipital electrode, *Figure 2C/D*). Maximal field intensities for the mid-line configuration (*Figure 1*) are 0.28 V/m ± 0.06 V/m across the ten subjects (*Figure 6A*). For an intensity of 2 mA typically used in clinical trials this would result in fields of 0.56 V/m. For other stimulation configurations on Subject P014 with electrodes away from the mid-line (*Figure 2A*, which is more typical for clinical interventions), maximum intensity on cortex for 1 mA stimulation was 0.25 V/m and 0.10 V/m at the fronto-lateral and lateral occipital locations respectively. We also model other electrode configurations commonly used clinical trials, e.g., M1–SO (*Figure 6A*). For the three configurations tested here, maximum field intensities in cortical locations was 0.38 V/m ± 0.09 V/m, again for 1 mA. To provide a sense of what intensities are reached in more extended areas (not just at the 1 mm$^3$ voxel with largest value) *Figure 6A* also reports the 95th percentile (0.14 V/m ± 0.02 V/m), following previous practice (*Parazzini et al., 2011*; *Alam et al., 2016*).

To estimate a margin of error on these values, consider the spread of the measured values shown in *Figure 5E*. For a given predicted electric field intensity, measurements vary by approximately 0.05 V/m. Maximal recorded values at intracranial electrode locations are about 50% of the maximal predicted values across the entire brain (compare *Figure 5F* with *Figure 6A*). Assuming that prediction accuracy scales linearly with predicted electric field intensity, this suggests that prediction error for maximal field intensities is in the order of 0.10 V/m.

The configuration with both stimulation electrodes on the mid-line (*Figure 1*) is particular in that significant current is shunted through the highly conducting CSF in the inter-hemispheric fissure. As a result, a relatively strong electric field may be produced in deep brain areas (e.g. peri-ventricular white matter and anterior cingulate, see *Figure 2D* second row). By measuring distance from the center of the brain ([0,0,0] in MNI coordinates) we can chart the dependence of electric field magnitude with depth for each location in the model brain (*Figure 6B*). The maximal values at a given depth summarized for all subjects and montages indicate that deep brain areas may experience electric fields that are comparable in magnitude to the cortical surface (*Figure 6C*, e.g.

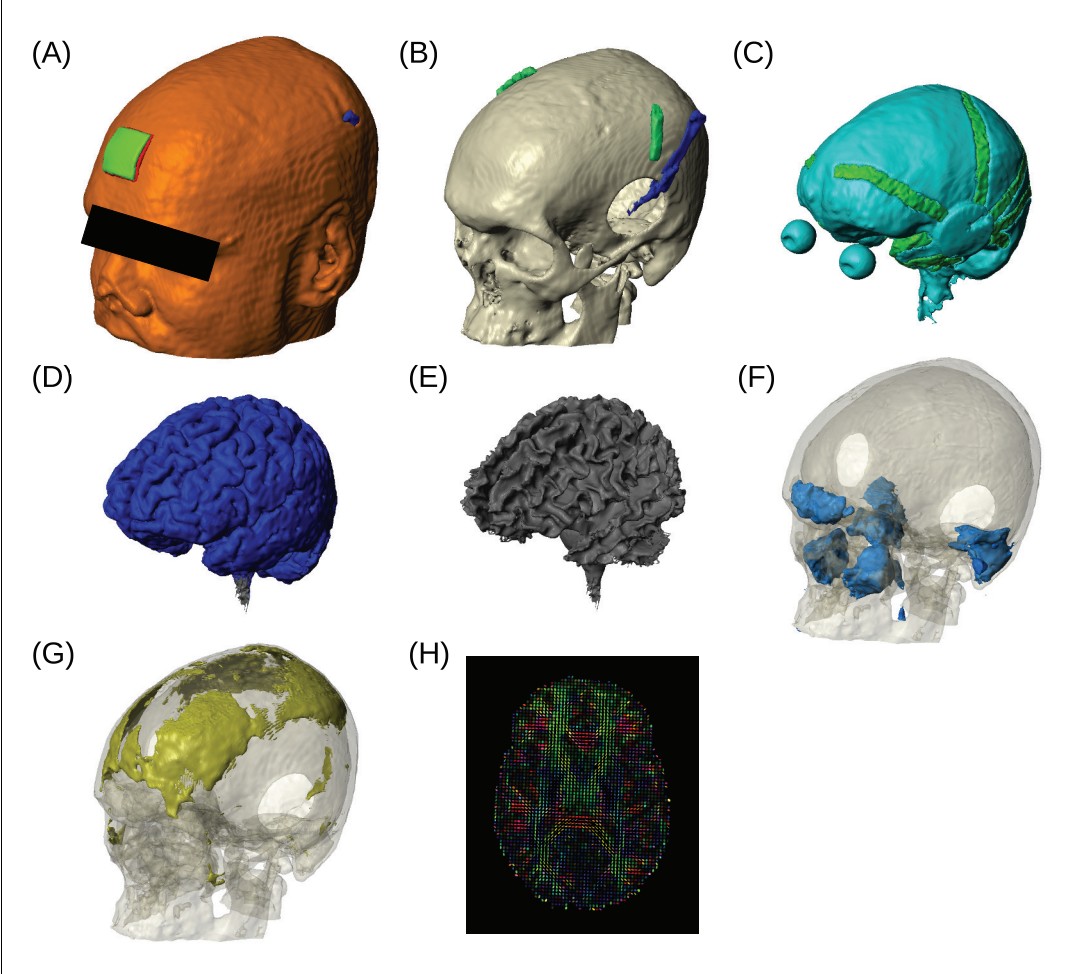

**Figure 4.** Example of realistic model for Subject P06. Each patient's detailed anatomy was obtained by segmenting T1-weighted MR images into six tissue types: scalp, skull, CSF, gray matter, white matter, and air. Additionally, to capture the surgical details we modeled the craniotomy, cortical strips and depth electrodes as well as the subgaleal electrodes. Finite element models were built and solved to compute voltages and electric fields throughout the head. (**A**) Scalp, with stimulating pad electrode; configuration used here is the same as shown in *Figure 1*. (**B**) Skull, note the Jackson-Pratt Drain (blue), the subgaleal electrodes (green) and the craniotomy. (**C**) CSF, with the geometry of intracranial electrode strips. Craniotomy site was assumed to be filled with CSF. (**D**) Gray matter. (**E**) White matter. (**F**) Air cavities. (**G**) Spongy bone inside the skull. (**H**) Diffusion tensor distribution in one brain slice.

approximately midway between the center of the brain and the cortical surface (normalized distance 0.25) field magnitude were 0.21 ± 0.04 V/m for these 10 subjects with the mid-line montage).

## Model calibration

In this and the following section we use the empirical voltage measurements to assess the validity of various modeling choices commonly used in the literature. First, conductivity values reported in the literature used in existing models appear to give similar estimate of the electric field magnitudes (see values in 'Conductivity optimization'). For example, measured electric field magnitudes in Subject P03 were close to the predicted values, as indicated by the slope $s = 0.72$ of the best linear fit of the measured versus predicted field projections shown in *Figure 7B*. This is true for all subjects (for voltage: $s = 0.83 \pm 0.24$; for field: $s = 0.68 \pm 0.21$. *Figure 8C/D* under 'literature'). The conductivity values typically used in the literature appear to correctly estimate the amount of current that crosses the skull and the amount shunted through the scalp. We calibrated the models by adjusting conductivity values for each individual model with the goal of minimizing the mean square error between predicted and measured field projections (see 'Conductivity optimization'; example of a

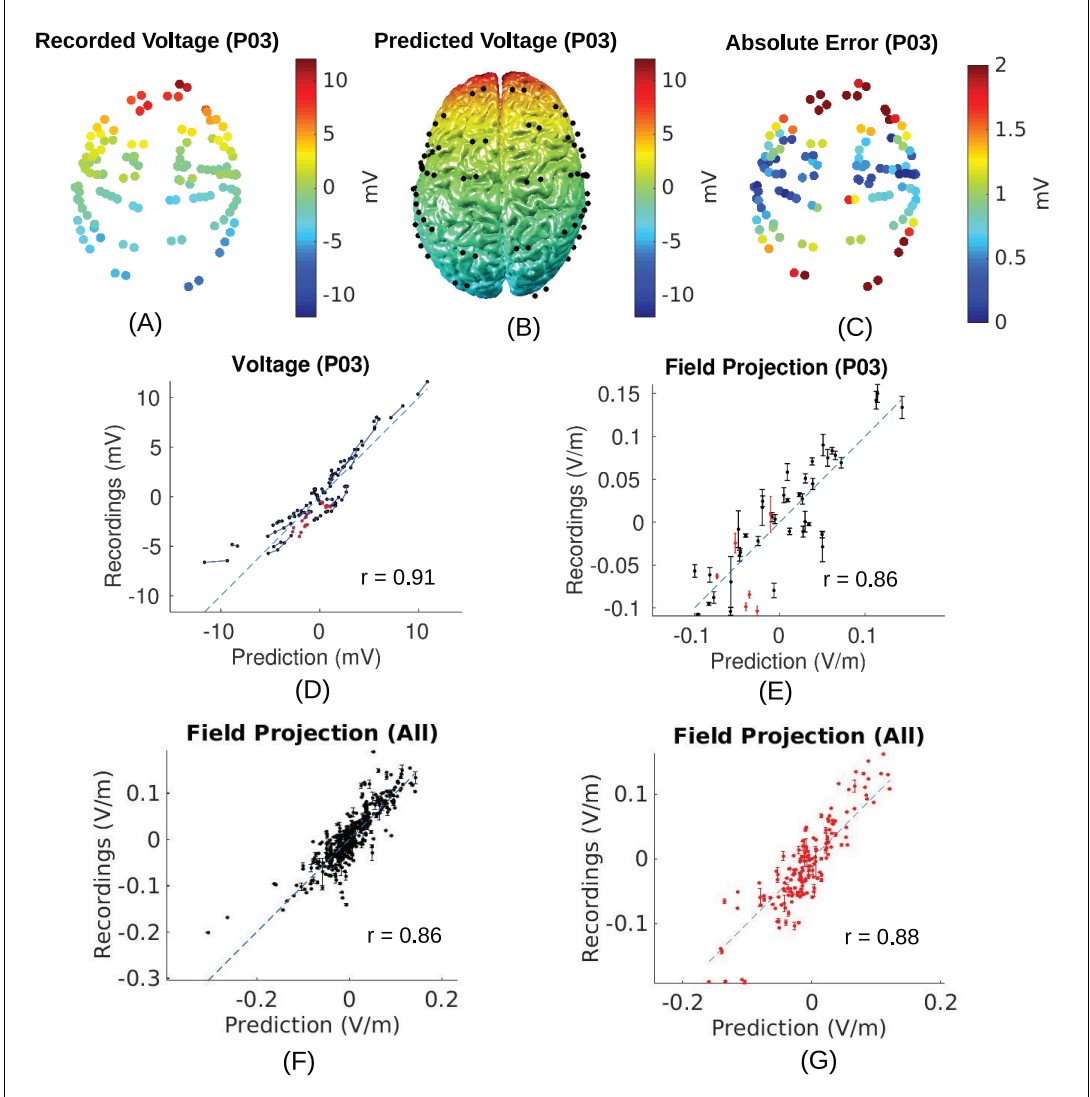

**Figure 5.** Voltage and electric field for measurements and model. All values are calibrated to 1 mA stimulation. (A) False-color representation of measured voltages for patient P03. (B) Voltages from the corresponding individualized model across the cortical surface. (C) Absolute voltage difference between recording and model predictions. (D) Comparison of recorded voltages with values predicted by the individualized model for P03. Each point in the scatter plot represents an intracranial electrode as shown in (A), with black indicating cortical surface electrodes and red representing depth electrodes (mostly targeting hippocampus). (E) Projected electric field is measured in the direction of nearby electrodes (pairs connected by blue lines in (D)), and is calculated as the voltage difference divided by the distance between the two electrodes. Error bar at each point indicates the standard variation of the measured electric field at the corresponding electrode as shown in *Figure 3C*). (F) Projected electric field for cortical surface recordings and corresponding model predictions combining all the subjects. (G) Same as (F) showing all the depth electrodes.

The following source data is available for figure 5:

**Source data 1.** Animated 3D renderings of the recorded and model-predicted voltages for each subject, and the absolute difference between the two.

best fit for P03 is shown in *Figure 5E*). The best fitted conductivity values vary across individuals (*Figure 8E–G*). Generally the fitting procedure showed that the parameters are not tightly bounded by the data. We estimated the uncertainty of the optimal conductivities by analyzing the sensitivity of the fitting criterion to small changes in these parameters (see 'Conductivity optimization' and *Figure 8—figure supplement 1*).

The median of the optimal conductivities are 0.03 ± 0.01 S/m for bone, 0.29 ± 0.10 S/m for skin, 0.38 ± 0.05 S/m for white matter and 0.82 S/m for gray matter (*Figure 8E–G*; ± indicates the median

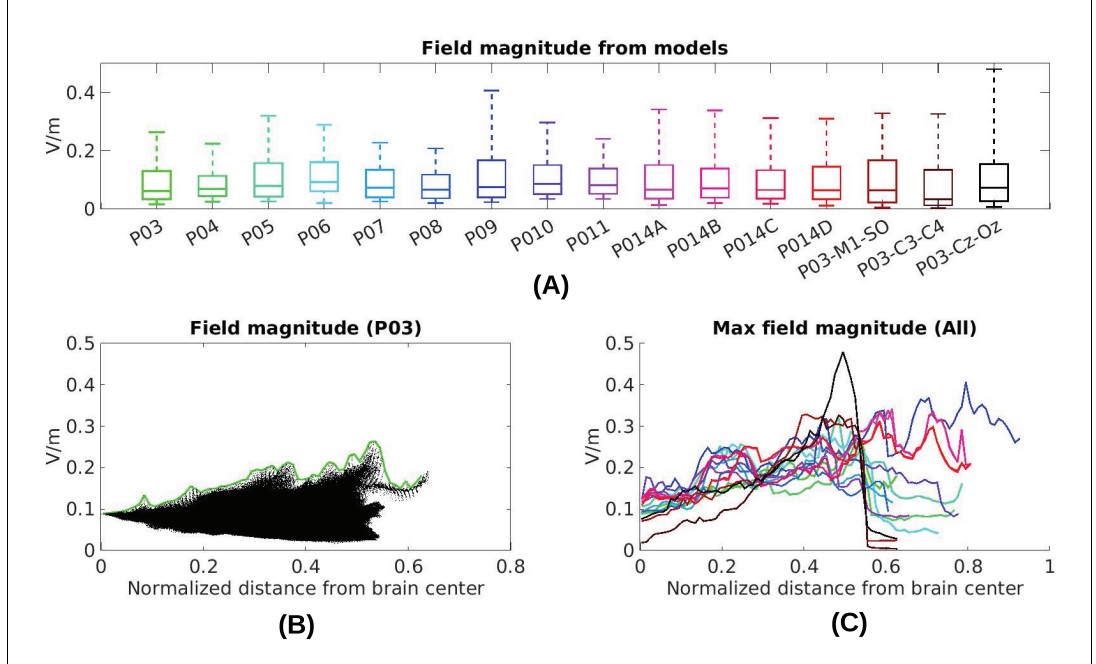

**Figure 6.** Electric field predicted with individually calibrated models under 1 mA stimulation. (**A**) Summary of electric field magnitudes for all subjects. The four different configurations of stimulation electrodes in Subject P014 are indicated as P014A–P014D. Also shown are values for a few stimulation montages commonly used in clinical trials simulated for Subject P03 (M1–SO, C3–C4, Cz–Oz). Whiskers indicate the maximal and minimal values of electric field magnitudes observed across the entire brain, and box indicates the 5% and 95% percentile across locations. Line inside the box indicates median value. (**B**) Electric field magnitudes as a function of depth, measured as the distance from the origin of the MNI coordinate system and normalized by diameter of the brain. Maximal field value is achieved at the cortical surface, which is approximately at distance of 0.55 (distance was divided by brain diameter in each MNI dimension). Locations exceeding 0.55 indicate mostly brain stem and cerebellum. Maximal value for each depth is indicated in green. (**C**) Summary of maximum for each of the 10 subjects and montages shown in (**A**) as a function of depth.

of the estimation uncertainty across subjects). These differ from the literature values (bone: 0.01 S/m; skin: 0.465 S/m; white: 0.126 S/m; gray: 0.276 S/m), but are largely in the same proportions. During optimization CSF was kept constant at 1.65 S/m. Compared to models using literature conductivities, the models with median values across subjects give significantly better accuracy in terms of predicting the electric field distribution (pairwise t-test: $t(12) = 2.36$, $p = 0.04$, *Figure 8B*), but the magnitude is not significantly different ($t(12) = 1.51$, $p = 0.16$, *Figure 8D*). Note that for magnitude, the t-test was performed on the absolute distance of slope $s$ to slope of 1.

## Relative merit of various model refinements

Since the introduction of detailed anatomical models at 1 mm³ resolution (*Datta et al., 2009*) there have been significant efforts to improve computational current-flow models by including additional tissue compartments (i.e. CSF, bone layers, and white matter). However, there is an ongoing debate as to the relative merits of each of these model refinements (see 'Guidelines for modeling'). To provide some guidelines to future modeling efforts we built models incorporating various refinements proposed in the literature (*Figure 9*) and compared model predictions to the empirical data in terms of the spatial distribution of projected electric fields (correlation metric reported in *Figure 8B*). To test whether individual anatomy is important for current-flow modeling, we compared recordings from one subject to predictions from other subjects. Specifically, for each subject (e.g. P03), the recordings were compared to model predictions obtained from other subjects (e.g. P04–P014) at co-registered electrodes locations. This was done for all the 10 subjects, each one comparing with the other 9 (a total of 90 comparisons). Correlation of recordings with these predictions using a different head model is significantly worse as compared to predicting values from the correct individual model (t-test on the difference between correlations: $t(89) = 5.60$, $p = 10^{-7}$). This confirms that

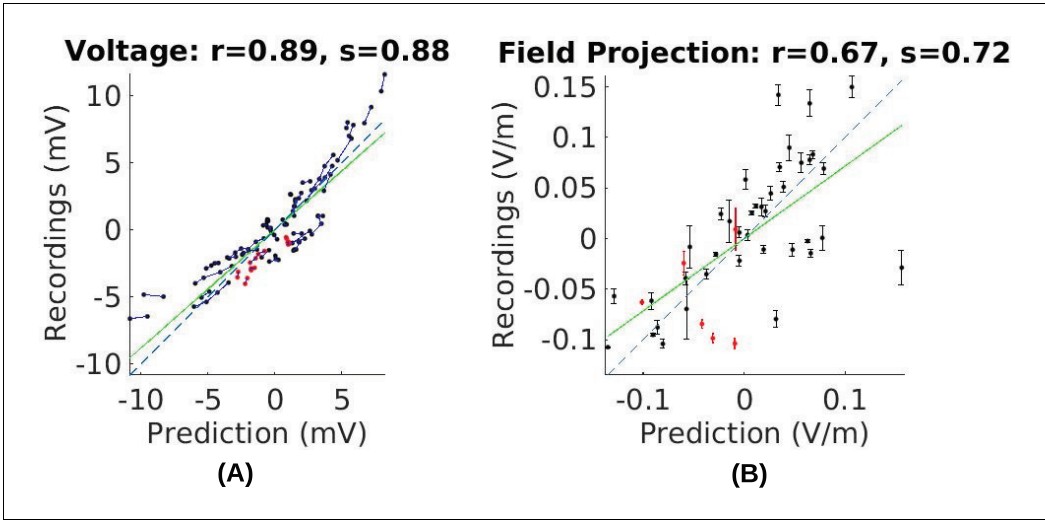

**Figure 7.** Comparison of recorded values with model predictions using literature conductivity values for Subject P03 scaled to 1 mA. Points falling on the dashed blue line represent perfect prediction (slope *s* = 1). The literature values gives close estimates of electric field magnitude (measurements are 72% of predicted values, *s* = 0.72, green line). Skin, skull and brain conductivities are optimized to minimize prediction error for field projections (i.e. minimize mean square distance from dashed line in panel (**B**)) which corrects this magnitude mismatch, and is shown in *Figure 5E*.

modeling individual anatomy is important. Many have argued that it is important to incorporate the entire head anatomy down to the neck. Since the MRI of most patients is truncated at the base of the skull, we extended the field of view (FOV) using a standard head that captures the average anatomy of the lower head (see 'General procedure'). This extended FOV gives significantly better predictions as compared to the original FOV, in terms of the electric field magnitudes ($t(10) = 6.17$, $p = 10^{-4}$) and electric field distributions ($t(10) = 2.63$, $p = 0.03$; *Figure 9B*, comparing RMcut vs. RM). Note that P04 and P06 were not included in this comparison as the FOV on these two subjects could not be extended (see 'General procedure'). We next tested the importance of CSF by removing the CSF compartment from the intact model – a model capturing the individual anatomy prior to craniotomy and electrode implantation. This did not significantly affect the distribution of estimated electric fields (*Figure 9B*, comparing correlations for IM vs. IM-CSF, $t(12) = 1.78$, $p = 0.10$), but did deteriorate the accuracy in predicting the magnitude of electric fields (comparing slopes, $t(12) = 4.39$, $p = 10^{-4}$). Incorporating heterogeneous conductivities for various bone compartments does not appear to provide a reliable improvement on magnitude estimates (RM vs. RM + 3skull: $t(12) = 1.72$, $p = 0.11$), and it seems to give worse prediction on the electric field distribution (*Figure 9B*, RM vs. RM + 3skull: $t(12) = 2.35$, $p = 0.04$). For some patients we had diffusion tensor imaging (DTI) data available (P06, P07 and P09). Previous modeling studies have advocated including such data to capture anisotropic conductivities. *Figure 9* shows the result of incorporating DTI information to the realistic model (RM+DTI) including different ways of converting DTI information into conductivity (RM+DTI/VN, RM+DTI/VC, RM+DTI/EIT; see 'Model categories' and *Table 1*). Three data points are not sufficient for a statistical comparison, but there does not appear to be an obvious advantage to using DTI. Note that all model comparisons were performed using the same fixed conductivity values (from the literature) to avoid biasing the comparisons with optimized parameters. Additionally, model comparisons focused on the accuracy of the spatial distributions, as captured by the correlation values, as they are not strongly impacted by the specific choice of conductivities (*Figure 8A/B*).

## Discussion

We set out to conclusively answer questions related to the electric field intensities that can be achieved in the human brain *in vivo* with transcranial electric stimulation. We also tested whether

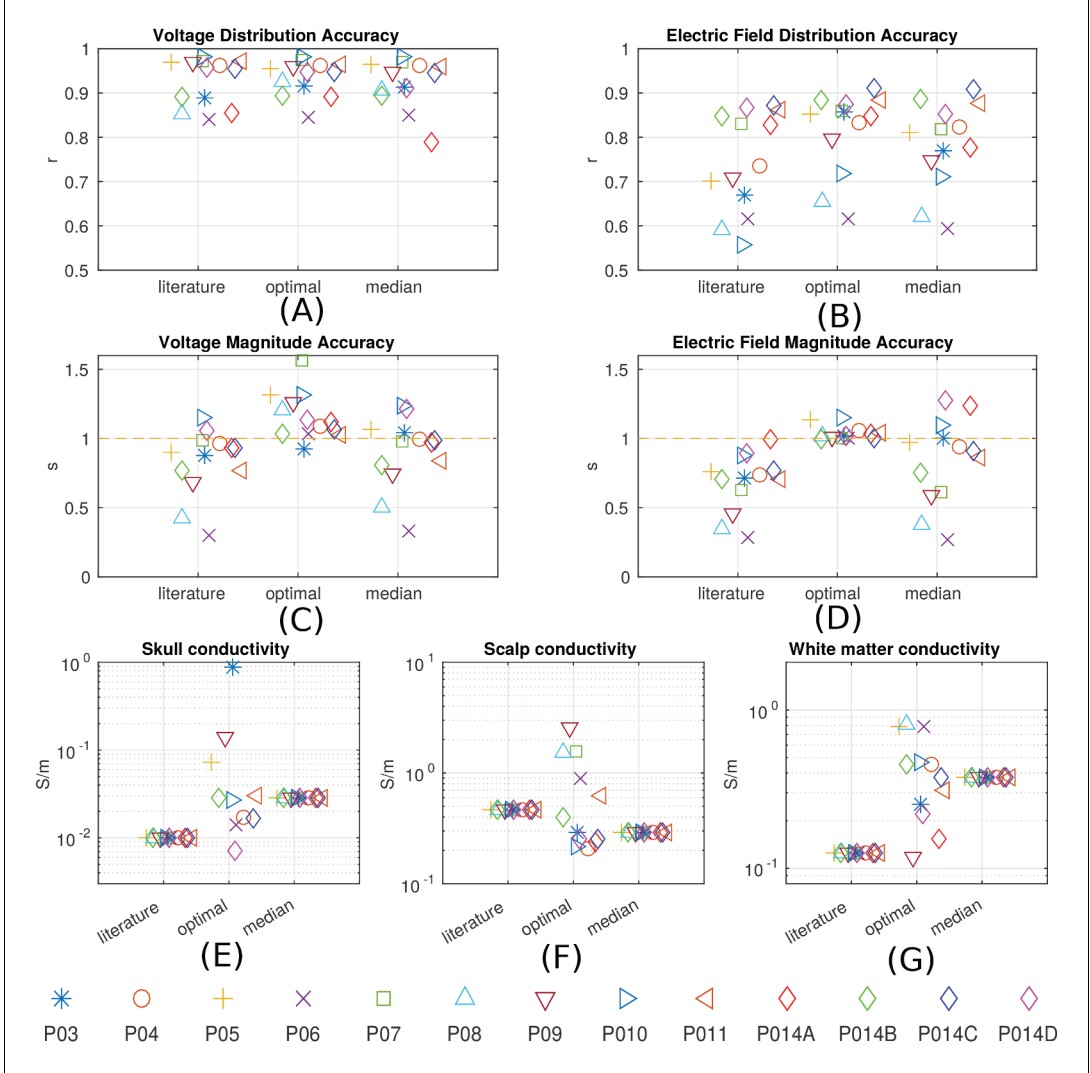

**Figure 8.** Prediction accuracy for models using various conductivity choices. (A, B) Correlation indicates the accuracy of the spatial distribution. (C, D) Slope indicates the accuracy of the magnitude estimate. Results are shown for three categories of models: models using literature conductivities (literature), models using individually optimized conductivities for skull, scalp and brain to provide best fit to the measured electric fields in each subject (optimal), and models with the median of the optimal conductivities (median of P03–P011 and P014). Each subject is represented by a different symbol as indicated by the legend on the bottom of the figure. P014A–P014D represent the four different configurations of stimulation electrodes in P014. Panels (E) – (G) summarize different optimal conductivities for different individuals.

The following figure supplement is available for figure 8:

**Figure supplement 1.** Estimation of the sensitivity of the fitting procedure to small variations in the conductivity values.

computational current-flow models accurately predict spatial distributions of electric fields, and which modeling choices are most warranted. Our main finding is that the intensities of electric field reach 0.8 V/m when using 2 mA transcranially. This is close to previous predictions using computational models (**Datta et al., 2009**). Peak intensities are achieved underneath the stimulation electrodes, but also in deep midline structures such as the anterior cingulate and the peri-ventricular white matter for the specific configurations tested here. We calibrated models by adjusting skull, scalp and brain conductivity in individual subjects to provide correct electric field magnitudes. We find that individualized models provide predictions of the spatial distribution of currents with an accuracy of $r = 0.86$ for cortical electrodes and $r = 0.88$ for depth electrodes when pooling data across all

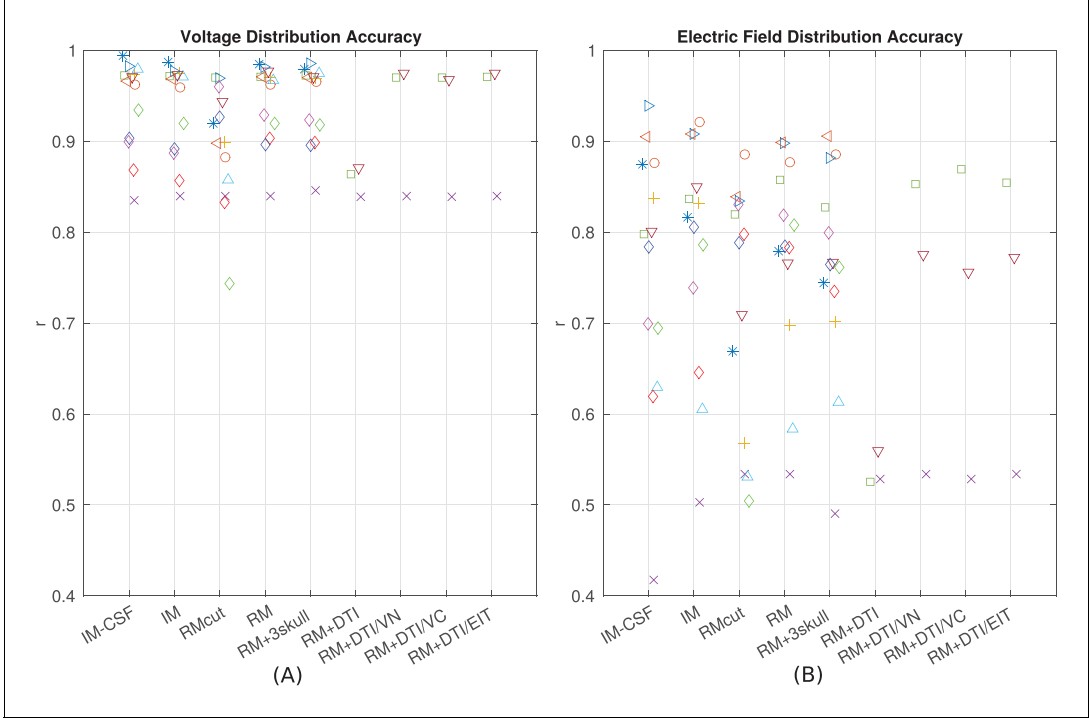

**Figure 9.** Performance of various modeling approaches. IM-CSF: This 'intact model' is based on the pre-surgical MRI and does not include craniotomy, recording electrodes, etc., and does not model CSF either; IM: intact model including CSF; RMcut: realistic model with all details as shown in *Figure 4A–F*, but truncated at the bottom of the skull due to the limited FOV of the clinical MRI scans; RM: realistic model with an extended FOV including the lower head and neck based on a standard head model; RM + 3skull: realistic model including 3-compartment skull as shown in *Figure 4G*; RM+DTI: realistic model including DTI as shown in *Figure 4H*. Four different ways to convert DTI ellipsoids into estimated anisotropic conductivity values were tested: direct method (DTI), volume normalized (DTI/VN), volume constrained (DTI/VC), and equivalent isotropic trace (DTI/ EIT). What is demonstrated is that truncated head models may deteriorate prediction accuracy, and models accounting for CSF, multiple skull compartments or white matter tracts do not significantly improve model accuracy.

subjects. These models capture individual anatomy for brain, CSF, skull, air cavities and skin at 1 mm$^3$ resolution. Including details such as anisotropic white matter and inhomogeneous bone compartments does not improve prediction performance. However, extending the FOV to include the entire head and neck significantly improves prediction accuracy.

**Table 1.** Models with different complexities

| Model | Details | Abbreviation |
|---|---|---|
| intact model without CSF | gray, white, skull, scalp, air, stim electrodes | IM-CSF |
| intact model | gray, white, CSF, skull, scalp, air, stim electrodes | IM |
| realistic model | intact model with craniotomy and surgical instrument | RM |
| realistic model with limited FOV | same as realistic model except truncated at the bottom of the skull | RMcut |
| realistic model with inhomogeneous skull | skull is modeled as 3-layered structure | RM + 3skull |
| realistic model with anisotropic brain derived from DTI data | direct mapping | RM+DTI |
| | volume normalized | RM+DTI/VN |
| | volume constrained | RM+DTI/VC |
| | equivalent isotropic trace | RM+DTI/EIT |

## Stimulation intensity

There is much debate as to whether electric fields in the brain with transcranial electric stimulation at conventional current intensities ($\leq$2 mA) are strong enough to have a physiological effect. Our study provide a definitive answer to the question of what field intensities can be achieved in human *in vivo*. Maximal stimulation intensity is difficult to measure directly because the recording electrodes are placed based on clinical considerations, and are thus not arranged where maximal intensities are expected. Subdural electrodes allow measurement of the electric field only tangentially to the cortical surface, whereas strongest field magnitudes underneath the stimulating electrodes are oriented orthogonally to the cortical surface. We thus used the validated and calibrated computational model to estimate the maximal electric field that can be achieved anywhere in the brain. Electric field magnitudes in cortex can be as high as 0.4 V/m for 1 mA stimulation current. For typical electrode configurations used in clinical trials maximal field intensity can reach 0.8 V/m when applying 2 mA. More extended areas can reach a value of 0.28 V/m (95th percentile) under 2 mA stimulation. For some electrode montages, current is shunted into deeper areas through highly conductive CSF with maximal values reaching 0.21 V/m. The present data is broadly in agreement with previous modeling estimates with maximal intensities around 1 V/m (e.g., *Datta et al., 2009*) and 95th percentile of 0.35 V/m (e.g., *Datta et al., 2012*). While physiological effects *in vitro* have been found for electric fields as low as 0.2 V/m (for entrainment of coherent gamma oscillations (*Reato et al., 2010*), other effects have required electric fields of at least 5 V/m (to entrain up/down state transitions (*Fröhlich and McCormick, 2010*); see *Reato et al. (2013)* for review on AC effects *in vitro*). Plasticity effects have been demonstrated only at higher field intensities of ~20 V/m (*Ranieri et al., 2012*; *Kronberg et al., 2017*). It is possible that network effects amplify electric field effects (*Reato et al., 2010*) and that larger neurons will be more strongly polarized (*Radman et al., 2009*) and thus, *in vivo* effects in human may be stronger than the effects demonstrated for animal *in vitro* experiments. Future *in vivo* animal work may shed light on this fundamental question of the required electric field intensities for various physiological effects (e.g., *Kar and Krekelberg, 2016*).

A recent study has reported the electric field produced by TES as measured by stereotactic EEG electrodes in two epilepsy surgical patients (*Opitz et al., 2016*). While the study does not leverage or evaluate accuracy of computational models, it does estimate maximal electric field to be 0.5 V/m with 1 mA TES. This closely approximates the intensity estimated from our current data (0.4 V/m for 1 mA stimulation). These results are in remarkably close agreement if one considers the differing placement of the stimulation electrodes. We made the deliberate choice of placing electrodes away from the surgical craniotomy to minimize current flow through the highly conducting skull defect. Previous modeling work suggests that electric field estimates do not significantly differ from those for normal anatomy when stimulating electrodes are distant from the skull defect (*Datta et al., 2010*). We confirmed this in our models, which show only minimal differences in field estimates when modeling the craniotomy as highly conducting CSF (in practice it is largely fluid-filled) versus entirely resistive (air filled). However, when electrodes are directly placed over the skull opening, as in *Opitz et al. (2016)*, one expects larger electric field intensities than normal as current flows directly into the skull. We thus feel that our estimate is closer to what one may expect in normal skull anatomy. *Opitz et al. (2016)* also measured voltages for different stimulation frequencies and found a 10% drop in voltage recording between 1–100 Hz. In our equipment we find a 25% drop which we attribute to a non-uniform gain of the equipment (in saline we measured a gain difference of 25% between 1 Hz and 100 Hz).

## Model validation

Early pioneering work attempted to validate spherical head models with measurements from a human half-skull suspended in a head-shaped receptacle filled with saline (*Rush and Driscoll, 1968*). Predictions of the spherical model were also compared to *in vivo* recordings from a monkey brain (*Hayes, 1950*), and scalp measurements in human (*Burger and Milaan, 1943*). In *Datta et al. (2013)* we compared an anatomically detailed model of a human head (of 1 mm$^3$ resolution) with voltages recorded on the scalp surface. But extrapolating the accuracy of model predictions from the scalp surface to the brain is problematic given the ambiguity as to the amount of current entering the skull. There are other indirect efforts to validate models by comparing them to the physiological effects, for instance, motor threshold *in vivo* in simian (*Lee et al., 2015*) and human (*Edwards et al.,*

*2013*; *Opitz et al., 2013*). *Bangera et al. (2010)* performed *in vivo* intracranial recordings from human, but the data were used to better predict EEG surface recordings, not for calibrating models of electric field with transcranial current stimulation. The most recent studies (*Opitz et al., 2016*; *Koessler et al., 2016*) involve similar *in vivo* intracranial recordings on human subjects and monkeys, but they do not provide a comparison and validation of current-flow models.

Accuracy of the models was assessed here in terms of the spatial distribution and the overall magnitude. Accuracy in magnitude was measured as the fraction between measured and predicted values (slope of a linear fit). After calibrating the models the overall magnitudes of electric field were predicted with a single set of parameters across all subjects without a bias ($s = 0.84 \pm 0.31$). Accuracy of the spatial distribution was measured as the correlation of measured and predicted values across recording locations. The individualized models at 1 mm$^3$ resolution provide predictions that correlated with measured electric field intensities ($r = 0.81 \pm 0.09$) across the ten subjects (*Figure 8B*). These numbers include optimization of conductivities to best match the recordings and may thus be overly optimistic. However, this bias is small as correlation is only minimally affected by the specific choice of conductivities. Indeed, when using a single set of values (median) we find similar performance ($r = 0.78 \pm 0.10$). Estimating spatial distribution of field magnitudes is important if one intends to use models to select the best electrode configuration to stimulate a particular brain region (*Dmochowski et al., 2011*; *Ruffini et al., 2014*; *Guler et al., 2016*). Many clinical trials and cognitive neuroscience experiments with TES aim to target a specific brain region, and many of the inferences that are drawn from such trials are predicated on having an accurate understanding of which areas are maximally stimulated, what polarity the stimulation has in specific sulci, and which areas are not significantly affected with a specific electrode montage.

## Guidelines for modeling

Current-flow modeling approaches range from spherical models (*Rush and Driscoll, 1969*; *Ferdjallah et al., 1996*; *Stecker, 2005*; *Miranda et al., 2006*; *Datta et al., 2008*; *Dmochowski et al., 2012*) to more realistic individualized models derived from MRI (*Wagner et al., 2004*; *Datta et al., 2009*; *Sadleir et al., 2010*; *Parazzini et al., 2011*; *Minhas et al., 2012*; *Datta et al., 2010*; *Wagner et al., 2007*). Individualized modeling implies the need to consider the anatomy of individual subjects (*Datta et al., 2011*; *2012*), in particular for patient populations that may have abnormal brain anatomy (*Datta et al., 2011*; *Dmochowski et al., 2013*). Various teams have also worked on automating segmentation and modeling for this purpose (*Acar and Makeig, 2010*; *Windhoff et al., 2013*; *Dannhauer et al., 2012*; *Huang et al., 2013*; *Huang and Parra, 2015*; *Thielscher et al., 2015*). We confirmed on the present data that modeling individualized anatomy indeed is beneficial in correctly predicting electric field distribution across space. Clinical MRI scans are routinely truncated at the base of the skull limiting the FOV that is available for modeling. By extending FOV with a standard head to capture the anatomy down to the neck we demonstrated that capturing current flows through the entire head significantly improved the modeling results. While most studies argue that the CSF and fluid-filled ventricles should be included in models (*Vorwerk et al., 2014*; *Wagner et al., 2014*; *Opitz et al., 2015*), others have argued that conventional 1 mm$^3$ MRI resolution cannot measure CSF space accurately and thus, it can be omitted (*Im et al., 2008*; *Park et al., 2011*; *Jung et al., 2013*). Our results indicate that including CSF into the model is important for correctly predicting the magnitude of achieved electric field in the brain.

Other model refinements aim to capture the heterogeneous skull layers (*Sadleir and Argibay, 2007*; *Dannhauer et al., 2011*; *Suh et al., 2012*; *Rampersad et al., 2013*; *Wagner et al., 2014*). *Wagner et al. (2014)* argues that modeling the skull as multiple compartments only has a meaningful effect when there is significant volume of cancelous bone along the current path. A complicating factor is that present segmentation algorithms do not automatically extract cancelous bone, making this modeling particularly labor intensive. In our results, manually segmenting a second bone compartment to account for cancelous bone did not significantly improve results. In addition, some have aimed to incorporate anisotropic conductivities using DTI data (*Windhoff et al., 2013*; *Suh et al., 2012*; *Rampersad et al., 2013*; *Wagner et al., 2014*). We could not evaluate the benefits of DTI modeling statistically, as we had only three subjects with DTI data. But for these three subjects there was no obvious benefit from adding DTI. This is consistent with previous modeling (*Wagner et al., 2014*), which shows limited effects of DTI-derived anisotropy on the estimates of electric field in cortex (most of our electrodes were in cortex). Additionally, there have been debates as to how exactly

the diffusion anisotropy (as measured with DTI) should be converted into anisotropy of the electrical conductivity (*Shahid et al., 2013*). In fact, the most recent *in vivo* recordings in humans suggest that white-matter electrical conductivity may actually be isotropic (*Koessler et al., 2016*), at least when measured at 50 kHz. Here we evaluated the various approaches proposed in the literature, and found no obvious performance difference between them.

The overall correlation of predicted and measured fields is 0.86 for cortical electrodes, and 0.88 for depth electrodes (*Figure 5F/G*). This indicates that despite our best efforts, there evidently remain some model inaccuracies. Simplifying tissue to have uniform conductivity is certainly a source of inaccuracy. Improving on this may require improvements in electrical impedance tomography. A source of uncertainty in our experiment is the placement of the stimulating electrodes. We know that small differences in placement significantly affect model predictions, yet the placement could not be conclusively established as the posterior electrode was occluded by bandages and we therefore only have rough anatomical landmarks that guided the placement. Future efforts may want to establish exact electrode placement using 3D localization devices. Other inaccuracies in the modeling may result from errors in segmentation, in particular thin structures such as CSF or temporal bone, which are in some locations below the 1 mm³ resolution of the model. Finally, openings in the eye socket such as optical foramen and orbital fissure are hard to establish and could serve to guide significant current for the present electrode montages. The models we tested here did not incorporate all the details that could conceivably affect current flow, and indeed, model accuracy could hopefully be improved from the currently reported values by adding such details. For instance, fatty tissue has been modeled by *Truong et al. (2013)*, who suggested that electric fields are altered in particular for obese individuals. Other details that may be important are muscle, arteries, sub-compartments of the eyes, and other anatomical details (*Sadleir et al., 2010*; *Parazzini et al., 2011*; *Mekonnen et al., 2012*). A general sense for the sensitivity of the model accuracy to errors in segmentation can be obtained by comparing the prediction performance of the intact models and the realistic models (IM vs. RM in *Figure 9*). Despite multiple refinements to the segmentation in RM, there was no consistent performance improvement across subjects. To allow for further testing of such refinements we have made the MRIs and recorded voltage values freely available to the public (at 10.6080/K0XW4GQ1).

## Electrical conductivity values of human tissues

Another conundrum in TES modeling is the electrical conductivities of the tissues used for modeling. Values commonly used in the literature are in fact the mean values across multiple references (*Wagner et al., 2004*). These conductivities are mostly measured at 10 Hz or higher frequencies (*Gabriel et al., 1996*; *Baumann et al., 1997*; *Oostendorp et al., 2000*; *Peters et al., 2001*; *Hoekema et al., 2003*). To date the data for the human head measured *in vivo* under direct current (0 Hz) are rather scarce and date back over sixty years (*Burger and Milaan, 1943*; *Freygang and Landau, 1955*). Just very recently new *in vivo* measures became available (*Koessler et al., 2016*) largely confirming the literature values for brain tissue, although these were measured at 50 kHz. Whether these literature values provide correct estimates of electric field magnitude in computational models of the human head has never been directly evaluated *in vivo*. Here we took a phenomenological approach and adjusted the conductivity of scalp, skull and brain to match the recorded fields at 1 Hz. The obtained conductivities are 'optimal' in the sense that they best calibrate the resulting electric field intensities inside the brain. Generally, the model fitting appears to be underconstrained as differing parameters gave similar goodness of fit (*Figure 8B*). We thus constrained the optimization to only three free parameters. The main goal of optimization was to allow some flexibility for the model to more accurately reflect the overall field magnitudes, which were incorrectly predicted by the conductivity values from the literature. The 'optimal' conductivities thus primarily serve to calibrate, on individual heads, the overall magnitude estimates. Once calibrated, these models were then used to estimate maximal field intensities in each head. However, we caution against interpreting these 'optimal' conductivity values as the true conductivity values of tissue. These best-fit conductivity values are free to account and compensate as best as possible for all sources of modeling simplifications and errors. For instance, the best conductivity obtained for 'skin' has to account for the value of many different soft tissues such as muscle, fat and nerves. This is also true for the skull conductivity, which has to reflect in a single number, different bone layers, which will vary in relative abundance across and within individuals. Also, the optimal conductivities may

depend on the location of the stimulation electrodes, segmentation errors, and other modeling choices. Thus, 'optimal' values differ considerably across subjects. Nonetheless, the median values we report here appear to be a good compromise for predicting field distributions when individual calibration is not possible.

To provide a sense of how sensitive model predictions are to choices of the conductivity value we computed the estimation accuracy for the best-fit values. The optimization criterion is fairly narrow around these values with estimation margins for skull, scalp, and white matter conductivities of one tenth to one quarter around the optimal values (*Figure 8—figure supplement 1A–C*). The model also appears to be sensitive to gray matter conductivity with a 10% change in accuracy for a 10% change in conductivity values. It is important to note that the model fitting process is strongly affected by the electric field magnitude, which is approximately inversely proportional to conductivity values. On the other hand, the spatial distribution of the predicted electric field is much less affected by the detailed choice of conductivities. Thus, it may not be surprising that optimal values differ across subjects by as much as one order of magnitude, yet a single median value provides reasonable accuracy in terms of spatial distributions across all subjects (*Figure 8B*).

## Distribution and stability of field measurements

For the purpose of targeting, the most important aspect of modeling is not the overall magnitude of electric fields but rather their spatial distribution. Targeting aims to select the electrode configuration that achieves highest intensity in one location while perhaps minimizing stimulation intensity elsewhere in the brain (*Dmochowski et al., 2011*; *Ruffini et al., 2014*; *Guler et al., 2016*). We attempted to provide error bars on the estimated electric fields by comparing repeated measures over an interval of approximately 15–30 min. We found relatively little variation in the voltage measurements (±0.11 mV, e.g., *Figure 3C*), and note that the variations we did observe were likely due to subject movement. Recent evidence and our own observations confirm that voltage measurements depend sensitively on subject position and even blood pulse (*Noury et al., 2016*). Both phenomena are readily explained by the shunting effect of the highly conducting CSF. Small variations with subject position have been demonstrated to affect the thickness of CSF layer and thus modulate the current flow between brain and scalp (*Rice et al., 2013*). Therefore, the variability of electric fields estimated here (with relatively quiescent subjects) likely underestimates the true variability occurring in practice. We conclude that with $r = 0.78 \pm 0.10$ the spatial distribution of electric field predicted by computational models is likely not far from the actual field distributions when taking into account their inherent variability across time, due to subject motion, position, blood pressure, electrode contact, etc. In this sense, the present study suggests that the current high-resolution model is a good foundation in order to select among different possible electrode configurations, and that model predictions with regard to stimulation focality are meaningful.

## Materials and methods

### Human subjects and intracranial recording setup

This study was performed in epilepsy patients undergoing surgical evaluation with intracranial EEG electrodes at New York University Medical Center (NYUMC). This protocol was approved by the NYUMC Institutional Review Board (IRB) and all patients provided written informed consent as indicated by the IRB. Eligible subjects were approached and consented generally within 24 hr prior to or after surgery (by author AL). We enrolled 12 subjects between December 2013 to June 2016 (P03–P014). Patients had to be over 18 years old, fluent in English, and able to provide informed consent or have an eligible consenting adult readily available. Subjects were excluded for cognitive impairment (IQ<70), a space-occupying intracranial lesion, frequent electro-clinical seizures in the 24 hr immediately after surgery (P012), and incomplete data (P013). Therefore, we have included 10 subjects in this paper (P03–P011, P014).

Voltages were recorded from implanted subdural electrodes (Ad-Tech Medical Instrument, Racine, WI) – the same electrodes used for intracranial EEG recordings. Electrodes were arranged as grid arrays (8 × 8 contacts), linear strips (1 × 8 or 12 contacts), or depth electrodes (1 × 8 or 12 contacts), or some combination thereof. Intracranial EEG signals were referenced to a two-contact subgaleal strip facing towards the skull near the craniotomy site. A similar two-contact strip screwed to

the skull was used for the instrument ground. Grid arrays and strips made from stainless steel and embedded in silastic sheets were implanted on the cortex. They have 2.3 mm diameter contacts with 10 mm center-center spacing. Depth electrodes made from platinum-iridium are inserted deep into the brain (around hippocampus). They have 2.4 mm contact length with 5 mm spacing. The decision to implant, electrode targeting, and the duration of invasive monitoring were determined solely on clinical grounds and without reference to this study. Subdural electrodes covered extensive portions of lateral and medial frontal, parietal, occipital, and temporal cortex of the left and/or right hemisphere (see *Figure 1*). In total we recorded from 1380 intracranial electrodes: P03 (124), P04 (124), P05 (124), P06 (124), P07 (80), P08 (120), P09 (250), P010 (124), P011 (118), P014 (192). Recordings from grid, strip and depth electrode arrays were made using a NicoletOne C64 clinical amplifier (Natus Neurologics, Middleton, WI), bandpass filtered from 0.16–250 Hz and digitized at 512 Hz. The input impedance of the amplifier is above 100 MOhm, and electrode impedance was kept below 75 kOhm during the recordings. Variation in impedance of up to 50 kOhm across electrodes would contribute at most 0.05 percent error in the voltage measurements. Voltage variations due to varying impedance across electrodes are therefore negligible given other sources of variation, predominantly movement leading to altered current paths and transient noise.

An hour of the patients intracranial EEG recording prior to stimulation was reviewed by an epilepsy physician (AL) to exclude recent seizures. A physician (AL) performed a pre-stimulation clinical assessment (including assessment of the stimulation skin site and neurological examination) and was present at the bedside during the entire procedure to monitor for clinical safety. Simultaneously, the patients intracranial EEG recording was monitored in real-time at the bedside during stimulation for seizures (AL, DF).

Consented subjects had one 2 × 2 cm rubber electrode placed on the mid-forehead (Fpz), and a second similar electrode placed at the occiput (Oz) below the sterile dressing and distant from the surgical skull defect, and applied with conductive paste (*Figure 1*). One subject (P014) had electrode montages placed at differing locations (see *Figure 2A*). Subjects were then covered with a nickel-cadmium sheath to reduce environmental artifact during recording. Additional sources of environmental noise in the hospital setting were minimized when possible.

The stimulation protocol used the Neuroconn DC Stimulator Plus (NeuroConn, Germany), with a sinusoidal waveform, at variable frequencies and intensities between 0.25 mA and 1 mA, zero-to-peak amplitudes. Stimulation was performed during waking rest (eyes closed) and afternoon sleep. Maximum stimulation intensities were primarily determined by the threshold at which amplifier saturation occurred. Stimulation was immediately stopped in the event of an electrographic seizure. A repeat clinical assessment (including assessment of stimulation skin site and neurological examination) was performed after stimulation.

## Voltage and projected electric field measurements

Voltage recordings used the same electrodes as the intracranial EEG. Magnitude and sign of induced voltages were measured by fitting a sinusoid to the recordings during sinusoidal stimulation. This fitting is insensitive to any DC voltage bias of individual electrodes. Except for *Figure 3B*, all other magnitude estimates are based on recordings during 1 Hz stimulation. These magnitudes were then calibrated by dividing with the intensity of the stimulating current to correspond to 1 mA current injection. The output of this post-processing were plotted and manually inspected electrode by electrode. Those electrodes with strong 60 Hz interference and evident clipping (strong harmonics in the harmonic fit) were discarded and were not used for model validation. In total we were able to use 1205 electrodes among a total of 1380 electrodes: P03 (118/124), P04 (112/124), P05 (118/124), P06 (117/124), P07 (78/80), P08 (116/120), P09 (184/250), P010 (101/124), P011 (111/118), P014 (150/192). Mean and standard deviation per electrode were computed whenever more than one recording session was available (all except Subject P04 and the last two stimulation montages for P014).

The projected electric field, for both recordings and model predictions, was calculated in the direction of adjacent electrode pairs by subtracting voltage values and dividing by their distance resulting in a measure with unit of V/m. Since electric fields are non-uniform, only local estimates of projected fields were measured by taking the difference only between the closest electrodes. Other field directions and averages could be obtained by looking at more distant pairs, at the cost of making systematic errors due to non-uniformity. Thus, for each electrode, a pair was selected as the

closest electrode on the *same* grid/strip that is within a 10 mm radius for cortical electrodes, and 5 mm radius for depth electrodes. Note that with this operation we are only capturing the magnitude in the specific orientation of the electrode pair used, that is, the cosine with the field vector.

## Computational current-flow models

### General procedure

MRIs were acquired on all the patients prior to the surgical implantation of electrodes. Within 24 hr after the surgery, patients underwent a post-operative MRI to confirm the locations of the implanted electrodes as part of their routine surgical evaluation. The model was built for each patient in the voxel space of the post-op image. However, this image usually has artifacts caused by the magnetic susceptibility of the implanted electrodes. Thus, we instead used the pre-op images to make the models. To this end, the pre-op MRI of each subject was registered and resampled into the voxel space of the post-op MRI.

The computational models were built following our previous work (*Huang et al., 2013*). Briefly, the registered and resliced pre-op image was segmented by the New Segment toolbox (*Ashburner and Friston, 2005*) in Statistical Parametric Mapping 8 (SPM8, Wellcome Trust Centre for Neuroimaging, London, UK) implemented in Matlab (R2013a, MathWorks, Natick, MA). Segmentation errors such as discontinuities in CSF and noisy voxels were corrected first by a customized Matlab script (*Huang et al., 2013*) and then by hand in an interactive segmentation software ScanIP (v4.2, Simpleware Ltd, Exeter, UK). The field of view (FOV) of the clinical MRI scans was extended down to the neck by applying SPM8 registration functions 'Coregister' (*Collignon et al., 1995*) and 'Normalise' (*Friston et al., 1995*) to a standard head (*Huang et al., 2013*), and then resampling the registered head and pasting it to the MRI of each individual patient. This process uses the soft-tissue as landmark for registration. It extrapolates reasonably well to align skull and soft-tissue in 8 of the 10 patients. For P04 and P06 we had to leave the FOV restricted to the original MRI scan. The 2 × 2 cm stimulation electrodes were manually created and positioned interactively into the model using ScanCAD module within ScanIP as follows. We took photographs of electrode placement in each patient, and placed the electrode in the model based on the positions shown in these pictures. However, this process was limited by the bandages which occluded the stimulating electrodes, in particular on the back of the head. There the electrode was placed halfway over the occipital ridge unless the photographs suggested otherwise. For each patient a finite element model (FEM, *Logan, 2007*) was generated from the segmentation data by the ScanFE module in ScanIP and then the electric potential distribution was solved by Abaqus (v6.11, SIMULIA, Providence, RI) under 1 A/m$^2$ current density injected into the frontal stimulation electrode (*Griffiths, 1999*). The resulting electric potential distribution was calibrated to correspond to 1 mA current injection.

The intracranial electrodes were localized and reconstructed from pre- and post-op MRIs using previously developed procedures at NYUMC (*Yang et al., 2012*), and their coordinates were registered into post-op images to read off the electric potentials in the brain predicted by the model. Specifically, the voltages on the FEM nodes at the brain that are closest to the electrodes were read out, and are ready to be compared against the actual measurements ('Validation criteria').

### Model categories

To assess how different parameters in the modeling process affect the model predictions, we built six categories of models for each subject. They are listed in order of increasing complexity in *Table 1*.

### Intact model (IM)

This is the model that is built on the immediate results from the segmentation process using the pre-op MRI, that is, model with normal head anatomy (gray matter (GM), white matter (WM), CSF, skull, scalp and air cavities). Anatomical alterations introduced by the surgery (e.g., craniotomy) are not included in this model.

### Realistic model (RM)

This is built upon the intact model, by further modeling all the structures introduced by the surgery. Specifically, the craniotomy was modeled by inserting a cylinder of the same diameter as the burr

hole into the skull and then intersecting the skull voxels. These voxels were given CSF conductivity and this area was covered by soft-tissue (skin conductivity). Additional instruments used during the surgery, such as the subgaleal electrodes and the Jackson-Pratt (JP) drain used to vacuum blood products, were modeled by manually drawing out the corresponding voxels from the post-op MRI. Since the intracranial electrode contacts are extremely small (the thickness is less than 1 mm in grid arrays and linear strips), we did not model them explicitly. Instead, we did model the geometry of the grid arrays and linear strips, by taking voxels from the CSF. To this end, customized Matlab script was written to generate cylinders along adjacent intracranial electrodes on the strips and arrays and then intersect with the CSF voxels. The insulated wires running through the scalp were not modeled as their diameter (<1 mm) is not expected to significantly affect the predicted current flow. *Figure 4A–F* shows a 3D rendering of such a realistic model for Subject P06. The realistic model also includes an extension of the field of view as described above.

## Realistic model truncated at the bottom of the skull (RMcut)

This is the same as the realistic model, except the lower head is cut off due to the limited FOV of the clinical MRI scans. Note for Subjects P04 and P06, RMcut is the same as RM, since for these two subjects we did not extend the FOV when building the models (see 'General procedure').

## Intact model without CSF (IM-CSF)

To test whether CSF is really important as claimed by many modeling studies (*Vorwerk et al., 2014*; *Wagner et al., 2014*), we removed the CSF in the intact models by combining it with the brain (eyeballs were combined with the scalp).

## Realistic model with inhomogeneous skull (RM-3skull)

Starting from the realistic models, we further differentiated the skull spongiosa and compacta. Specifically, manually determined thresholding was performed for each subject to segment out the spongy bone from the compact bone. See *Figure 4G* for an example. Note that we did not make the skull anisotropic, as three-isotropic-layered model was recommended by a modeling study and *ex vivo* measurement (*Sadleir and Argibay, 2007*). Thus, this skull model is inhomogenous and isotropic.

## Realistic model with anisotropic brain (RM+DTI)

Anisotropy (conductivity varying with direction) was added to the gray and white matter on three subjects for whom diffusion tensor imaging (DTI) scans were available (P06, P07, P09). The DTI scans were first corrected for non-linear susceptibility and eddy-current distortions with the FSL (*Smith et al., 2004*) 'topup' (*Andersson et al., 2003*) and 'eddy' (*Andersson and Sotiropoulos, 2016*) command-line tools respectively. The mean of the non-diffusion-weighted (b0) volumes was then used to create the DTI brain mask via the FSL 'BET' utility (*Smith, 2002*). Since the conductivity tensor is required to be positive semi-definite, the diffusion tensor was calculated for the brain voxels with a non-linear optimization relying on positive-definiteness constraints (*Jones and Basser, 2004*) as implemented in the Camino software (*Cook et al., 2006*). After this, to integrate the DTI data into the model, the fractional anisotropy (FA) of the diffusion tensor was computed and registered with an affine transformation to the voxel space of the post-op MRI (where the model is built). To circumvent the complexity of spatially transforming the diffusion tensors, the previous transform was applied to all the distortion-corrected diffusion-weighted volumes. The positive semi-definite diffusion tensor was then computed again from the transformed diffusion-weighted data, but within a different brain mask defined by the model. See *Figure 4H* for a slice view of the diffusion tensor of Subject P06. Each mesh node in the model was then associated to the nearest diffusion tensor. These diffusion tensors were then converted into conductivity tensors using one of four different algorithms used in the literature: direct mapping (*Tuch et al., 2001*); volume normalized (*Hallez et al., 2008*; *Güllmar et al., 2010*); volume constrained (*Wolters et al., 2006*; *Rullmann et al., 2009*); and equivalent isotropic trace (*Miranda et al., 2001*). Models with these anisotropic conductivities were also solved in Abaqus.

Note that when comparing across different categories of models listed above, literature conductivity values (see 'Conductivity optimization') were used when solving the models. This is to avoid any potential bias that may be introduced when using optimized conductivities ('Conductivity optimization').

## Validation criteria

For both the voltages and the projected electric fields, the model-predicted values and the recorded values (see 'Voltage and projected electric field measurements') were compared for each subject in terms of two criteria: (1) Pearson correlation coefficient $r$ between recorded and predicted values; (2) the slope $s$ of the best linear fit with predicted value as 'independent' and measurement as 'dependent' variables. The correlation captures how well the model predicts the distribution patterns of the electric potential regardless of a potential mismatch in overall magnitude. The slope measures how accurate the model estimates the absolute magnitudes. Since the ground reference for the recordings (the subgaleal electrodes) is different from the ground for the model (electrode placed on the back of the scalp surface), mean across electrodes was subtracted from the recordings and the model outputs before they were compared. Slope and correlation are both independent of an overall offset.

Pairwise t-test was carried out to compare how different conductivity values and different model types affect the prediction accuracies (*Figure 8* and *Figure 9*). Note that when comparing magnitude accuracy as indicated by the slope $s$, the t-test was performed on the absolute distance of $s$ to slope of 1.

## Conductivity optimization

We first assigned literature conductivities to the realistic models. The following values were used (in S/m): GM–0.276; WM–0.126; CSF–1.65; skull–0.01; scalp–0.465; air–$2.5 \times 10^{-14}$; gel–0.3; electrode–$5.9 \times 10^7$ (*Crille et al., 1922*; *Burger and Milaan, 1943*; *Freygang and Landau, 1955*; *Ranck, 1963*; *Hasted, 1973*; *Geddes, 1987*; *De Mercato and Garcia Sanchez, 1992*; *Gabriel, 1996*; *Akhtari et al., 2002*). For realistic models with inhomogeneous skull: spongiosa: 0.02865 S/m; compacta: 0.0064 S/m (*Akhtari et al., 2002*). The JP Drain and intracranial electrode grids and/or strips were made almost insulated ($<10^{-14}$ S/m).

We note that in the literature cited above a range of values are reported, all of which have been measured at 10 Hz or higher frequencies (up to 1 GHz, in S/m): skull: $0.0028 \sim 0.08$ (*Oostendorp et al., 2000*; *Akhtari et al., 2002*; *Hoekema et al., 2003*); scalp: $0.0003 \sim 1.0$ (*Gabriel et al., 1996*); WM: $0.1 \sim 1.0$ (*Gabriel et al., 1996*); GM: $0.016 \sim 1.0$ (*Freygang and Landau, 1955*; *Gabriel et al., 1996*; *Peters et al., 2001*; *Logothetis et al., 2007*).

Secondly, the optimal conductivity for each patient was obtained by fitting the model outputs to the recordings. Formally, the following optimization problem was solved:

$$\sigma^* = \underset{\sigma}{\arg\min} f(\sigma), \quad \text{where} f(\sigma) = \frac{1}{M} \sum_{i=1}^{M} \left[ E_i - \hat{E}_i(\sigma) \right]^2. \tag{1}$$

Here $E_i$ is the projected electric field on electrode $i$ calculated from recorded voltage, and $\hat{E}_i(\sigma)$ is the corresponding model-predicted value, which depends on the conductivity $\sigma$. $M$ is the total number of intracranial electrodes. Mean across electrodes was subtracted from both recordings and model predictions before evaluating the cost function. A general purpose optimization algorithm – Pattern Search (*Audet and Dennis, 2002*), available in Matlab – was used to solve this optimization. This procedure was carried out only on the realistic model for each subject individually. In fitting conductivity parameters we found that the values are generally under-constrained (very different conductivity values can give similar goodness of fit). We therefore limited the fitting to adjusting scalp, skull, and brain conductivities, but fixed CSF (1.65 S/m) and maintained the ratio between gray and white matter constant (=2.19) reflecting the values reported in the literature. Thus there are only three free parameters in the optimization, which is also beneficial in order to keep computing times reasonable. Each model evaluation took approximately 30 min, and optimization was terminated after 300 evaluations of the Pattern Search algorithm. Note that the conductivities of the tissues are optimized jointly, that is, they are adjusted simultaneously per evaluation of the cost function.

To estimate the uncertainty in this parameter estimation we computed the variance of the estimator based on the Cramér-Rao bound (*Radhakrishna Rao, 1945*; *Cramér, 1999*). This was done separately for each conductivity value, which gives a loose lower bound for the estimation error, that is,

$$\text{var}(\sigma) \geqslant \frac{f(\sigma^*)}{Ma}, \tag{2}$$

where $\mathrm{var}(\sigma)$ is the estimation variance of the conductivity, and $f(\sigma^*)$ is the residual error (*Equation 1*). Here $a$ is the parameter of a quadratic fit to the cost function around the optimal value. The expression to the right is the inverse of the Fisher information, estimated numerically. For parameters that were fixed to the literature values (gray matter and CSF), we similarly solved the model for close-by values (±10%) and report the sensitivity of the normalized error to this 10% fluctuation. All values are shown in *Figure 8—figure supplement 1*.

Finally, we also tested the median of the optimal conductivity across all subjects (P03–P011, and P014).

## Acknowledgements

This work was supported in part by NIH grants R44NS092144, R01MH092926, R41NS076123, R01MH107396. The authors would like to thank Hetince Zhao for her dedicated work in manually improving the automatic segmentation results and building most of the realistic models. We also would like to thank Preet Minhas for her initial pilot efforts to record intracranial EEG during electric stimulation on P01 and P02.

## Additional information

### Competing interests

MB, LCP: Has significant interest in Soterix Medical Inc. which commercializes hardware and software for TES. He is listed as inventors on patents (U.S. Patent application No.13/264,142) related to TES. The other authors declare that no competing interests exist.

### Funding

| Funder | Grant reference number | Author |
|---|---|---|
| National Institute of Neurological Disorders and Stroke | | Yu Huang<br>Belen Lafon<br>Michael Dayan<br>Marom Bikson<br>Lucas C Parra |
| National Institute of Mental Health | | Yu Huang<br>Belen Lafon<br>Michael Dayan<br>Marom Bikson<br>Lucas C Parra |
| National Institutes of Health | R44NS092144 | Yu Huang<br>Lucas C Parra |
| National Institutes of Health | R01MH107396 | Anli A Liu<br>Daniel Friedman<br>Xiuyuan Wang<br>Werner K Doyle<br>Orrin Devinsky<br>Lucas C Parra |
| National Institutes of Health | R01MH092926 | Lucas C Parra |
| National Institutes of Health | R41NS076123 | Lucas C Parra |

The funders had no role in study design, data collection and interpretation, or the decision to submit the work for publication.

### Author contributions

YH, Formal analysis, Methodology, Writing—original draft, Writing—review and editing; AAL, Conceptualization, Data curation, Methodology, Writing—original draft, Writing—review and editing; BL, DF, Data curation; MD, XW, Formal analysis; MB, OD, Writing—review and editing; WKD, Data curation, Methodology; LCP, Conceptualization, Data curation, Formal analysis, Writing—original draft, Writing—review and editing

## Author ORCIDs

Lucas C Parra, http://orcid.org/0000-0003-4667-816X

## Ethics

Human subjects: This study was performed at the New York University Medical Center (NYUMC). The protocol was approved by the NYUMC Institutional Review Board(IRB) and all patients provided written informed consent prior to their participation in the study. A physician was present at the bedside during the entire procedure to monitor for clinical safety.

## Additional files

### Major datasets

The following dataset was generated:

| Author(s) | Year | Dataset title | Dataset URL | Database, license, and accessibility information |
|---|---|---|---|---|
| Yu Huang, Anli A Liu, Belen Lafon, Daniel Friedman, Michael Dayan, Xiuyuan Wang, Marom Bikson, Orrin Devinsky, Lucas C Parra | 2016 | Recordings of electrical potentials in the in vivo human brain induced by transcranial electrical stimulation. | https://dx.doi.org/10.6080/K0XW4GQ1 | Publicly available at Collaborative Research in Computational Neuroscience - Data sharing (http://crcns.org/) |

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
