## [Decision Letter]

Thank you for submitting your article "Measurements and models of electric fields in the in vivo human brain during transcranial electric stimulation" for consideration by *eLife*. Your article has been reviewed by three peer reviewers, and the evaluation has been overseen by a Reviewing Editor, Richard Ivry, and Sabine Kastner as the Senior Editor. The following individual involved in review of your submission has agreed to reveal his identity: Angel Peterchev (Reviewer #1).

The reviewers have discussed the reviews with one another and the Reviewing Editor has drafted this decision to help you prepare a revised submission.

Summary:

Thank you for submitting your work entitled "Measurements and models of electric fields in the in vivo human brain during transcranial electric stimulation" for consideration at *eLife*. As one of the reviewers noted, "This manuscript is of exceptional importance as it provides unique, much needed, and highly sophisticated measurement, modeling, and analysis of the electric field in the human brain due to transcranial electrical stimulation (tES)." The paper provides the first thorough validation of computational models of transcranial current stimulation. While there are concerns with the modeling approaches used to make electric field estimates, the measured fields in the vicinity of the electrodes provide a direct measure that are independent of the modeling. These estimates provide the most direct observations of tES induced E-field in the human brain, an extremely important contribution. The data set is substantial and, as indicated by the authors, will be made available publicly, a valuable resource for the community. In summary, the reviewers felt that the paper, when appropriately revised, will be a landmark report in this literature and of great use to the growing community interested in using, modeling, and optimizing this stimulation technology.

Essential revisions:

I) Concerns with the experimental results: These experimental results are very important. It is essential to make sure that the methods and analysis are clearly described and consider possible confounds, especially factors that might contribute to signal attenuation.

1) Along these lines, we want the revision to specify the input impedance of the amplifiers and the typical electrode impedance. It is stated that electrodes were considered high impedance if there was strong 60 Hz noise on these channels, but it is unclear if this approach eliminated electrodes with low enough impedance to prevent 60 Hz noise, but high enough impedance to cause an attenuation of the recorded voltages. Obviously, any unintended voltage attenuation, while not critical for intracranial EEG, can affect all estimates of E-field strength and tissue conductivities in this study. Furthermore, it is implied that there was clipping of the signal in some electrodes (data from them were not included in the analysis). During clipping did the input impedance of the recording amplifier decrease and could this produce current stimulus flow though the recording electrodes?

2) There is a preprint available on bioaRxiv that involves similar experiments on two patients and two NHP (http://biorxiv.org/content/early/2016/05/18/053892, Opitz et al.). The results reported in that preliminary report could be compared to the present work. Given the great interest we anticipate in your work and this other study, it would be useful to comment on the similarities and differences between the experimental results in two studies. Two issues that come up when comparing the studies are 1) possible frequency-dependent effects and the manner in which these were analyzed, and 2) differences in how electric fields are estimated. For example, for electrodes in the interior of the grid why not use all neighbors instead of just the closest one?

II) Concerns with the modeling work: We see the modeling work as a nice complement to the experimental data. We recognize that this work entails complex models, involving a number of assumptions. Below we summarize major issues to consider in revising this section. Most involve clarifying the methods and potential confounds of the models, and carefully qualifying the conclusions based on these models. In some cases, we would recommend conducting additional simulations and/or analysis since this will reduce qualifications you would have to place on the conclusions.

3) The study concludes that modeling anatomical detail such as the skull layers and the white matter anisotropy does not improve the model predictions. It seems, however, that the models were optimized only for the "realistic model" without the respective anatomical refinements. This should be clarified, and if this is indeed the case, the comparisons are biased since one side of the comparison is optimized and the other is not. Furthermore, the white matter conductivity anisotropy affects mostly the E-field in the white matter, while most of the voltage recordings were not in the white matter, which could affect efforts to optimize the anisotropic model fit to the data.

4) (also relevant to experimental results) The selected montage (e.g., inferior position), combined with the large inter-electrode distance increases the significance of conduction paths in the lower half of the head. These include regions with complex anatomy and conductivity profiles such as the eyes, orbits, optic nerve, neck, nasal cavity, etc., as well as the truncation of the head and neck (as seen in Figure 4). Furthermore, only a limited set of tissues are modeled, and other potentially significant ones are omitted such as muscle, fat, and sub-compartments of the eye. Discussion should be provided concerning the extent these regions affect the accuracy of the electric field simulation as well as the derived "optimal" conductivity of the scalp and skull.

5) The derived "optimal" conductivity values for scalp (mean = 3.61 V/m) are very high, being an order of magnitude above values in the literature and more than twice the measured conductivity of CSF and physiological saline (both ~ 1.6 S/m). It is unclear what physical mechanism could possibly confer such high conductivity. While this is acknowledged by the authors, this is an issue of concern. Since they are based on a fit of a complicated model with just two free parameters to a limited data set, it is entirely possible that the optimized conductivity values result from unrecognized limitations of the model and/or recordings. For example, truncation of the head and neck (as seen in Figure 4) or other anatomical inaccuracies in the lower portion of the head could artificially strengthen the modelled current flow in the top portion of the head, requiring higher scalp conductivity values to match the recorded potentials. Further, the optimized conductivity values may depend on the location of the tES electrodes. Adopting the "optimal" values from this study in general for tES (and potentially other) simulations may produce errors when the model context changes. We would like you to consider quantifying the impact of these model limitations; for example, test the impact of head truncation and reduction of the number of segmented tissue compartments in one of the full head and neck models. In addition, the revision should be explicit about these limitations and impact on inferences to be drawn from this work.

6) The paper attempts to solve two mathematical problems: (A) An inverse problem in which they the estimate conductivity (scalp and skull only) given the voltage measurements noted above and (B) A forward problem in which they calculate the scalar potential field and the electric field everywhere (or at least potentially so) within the head given the previously estimated conductivity and assumptions of their model (6 regions delineated by anatomical MRI with each region being of spatially constant conductivity). The solution of problem (A) is attempted through a minimization of error between measured and predicted values as given by Equation 1. The solution of problem (B) is obtained through the Abaqus software. For the estimates of the electric field to be meaningful, a reasonable accuracy estimate must be associated with the estimate. No such estimate is given. To do so would require at least the following: (i) A reasonable estimate of the error produced in results of problems (A) and (B) that is associated with assumption that the conductivity is spatially constant throughout each of the 6 segmented regions and (ii) A reasonable estimate of the error in the results of problems (A) and (B) associated the measurement and segmentation of the six regions. In addition, known instrumental error must be propagated through both problems as well.

7) Why aren't the correlation values between measured and predicted fields better than are reported here? If the solution to problem (A) is unique and if the voltage measurements are consistent with the quasi-static field equations, then why don't the results of problem (B) reproduce the measurements of voltages (albeit with some presumably small measurement error discrepancies)?

III) Concerns with the organization and clarity of the paper:

8) In general, there is concern with organization, clarity, precision, and to some extent discussion. The current organization is hard to follow. There are methods descriptions mixed into the Results section, while at the same time parameters reported on in Results (e.g. "s", "r", "t(n)" – the latter does not even seem to be explained in Methods)) without any description at that point of what they mean or any mention that they are described in Methods. Please review your manuscript from the reader point of view, rigorously stripping methods material from Results and inserting descriptive phrases and referrals to Methods in Results where needed. The statistics should be described more carefully in the Methods section. A number of experimental details are missing or not clear, including the specific criteria by which some patients were excluded and also by which electrodes were excluded within a given patient (since the latter is a very significant fraction of the total number of electrodes this seems particularly important) and what the range of electrodes was across all subjects (we are only given the total). Make sure the manuscript clearly describes when and why particular sections/graphs exclude some of the participants. There are a number of places were the grammar needs work, including instances where there is disagreement of subject and verb, missing prepositions, missing definite or indefinite articles, etc. Please review the manuscript to be as precise as possible (e.g., what does "as important as commonly thought" in the subsection “Relative merit of various model refinements”?).

[Editors' note: further revisions were requested prior to acceptance, as described below.]

Thank you for resubmitting your work entitled "Measurements and models of electric fields in the in vivo human brain during transcranial electric stimulation" for further consideration at *eLife*. Your revised article has been favorably evaluated by Sabine Kastner as the Senior Editor, Rich Ivry as the Reviewing Editor, and two reviewers.

In general, we are pleased with the extensive revisions you have provided. We continue to believe this is a very important paper, one that will be of considerable interest to the brain stimulation community.

There remains one important issue to be addressed. While you have provided E field predictions, the manuscript does not provide an associated estimate of the error term for these predictions. A complete response would address the following:

1) For conductivities that are allowed to vary in the current fitting procedure: Provide the range of conductivity values that work nearly as well as those used in the current version.

2) For conductivities that have been fixed in the fitting procedure: Provide a range of conductivity values that fall within reasonable estimates of error for those values.

3) A range of compartment segmentations taking into account a reasonable estimate of error for the accuracy of the segmentation methods.

We recognize that the work required for these three error estimates may be extensive, and also see considerable value in the current modeling work given that the electrical field estimates for each neighboring electrode contact pair correlate well with the measurements, indicating that the shape of the model solution is in agreement with the shape of the data. Moreover, the conductivities are in reasonable agreement with published values.

That said, at a minimum we would like to see #1 addressed, with a quantification of the uncertainty (confidence intervals) of the optimized conductivity values. Are the confidence intervals for the individually estimated conductivities available? This would provide information on the uncertainty in the optimization process, assuming the non-optimized parameters are fixed. In addition, you could propagate reasonable estimates of error in the model parameters through the predictive model to estimate error in the prediction.

We also ask that you consider #2 and #3, and in the ideal response, would again propogate the error estimates. The former could be done by varying the fixed conductivity parameters. As we see it, #3, while useful, might entail a lot of work and the added value would not justify the request. Nonetheless, if you have information of value here, this would certainly strengthen the modeling component of the paper. If you decide to focus on #1 only, then we would ask that, in the revision, you make explicit limitations with the current work in terms of establishing the accuracy of your approach since the paper would lack a thorough analysis of the estimates of error.

We apologize for the long turnaround time on this revision, but wanted to provide a clear request for what we anticipate will be a final revision, one that we should be able to act on in a very timely manner.

[Editors' note: further revisions were requested prior to acceptance, as described below.]

Thank you for resubmitting your work entitled "Measurements and models of electric fields in the in vivo human brain during transcranial electric stimulation" for further consideration at *eLife*. Your revised article has been favorably evaluated by Sabine Kastner as the Senior Editor, Rich Ivry as the Reviewing Editor, and two reviewers.

Just two revision requests to consider. I am confident that you will be able to address them quickly and then we can move to a final decision after the Reviewing Editor has a chance to take a look at the revision.

1) Reviewer 1 has two suggestions for improving Figure 5—figure supplement 1:a) In Figure 8—figure supplement 1, logarithmic spacing of the y axis could be considered given the wide range of the values (over a decade) and the seeming proportional behavior of the C-R bound error bars.b) In Figure 8—figure supplement 1, add lines connecting the two points for each subject in the corresponding color to allow easier visual inspection of the local sensitivity.

2) Reviewer 2 asks for more on your efforts to provide estimates of error. I am satisfied with what you have done on this. You could consider compiling the discussion of error estimation into a focused section to highlight this issue for readers. I will leave the final decision here up to you.

---

## [Author Response]

*Essential revisions:*

*I) Concerns with the experimental results: These experimental results are very important. It is essential to make sure that the methods and analysis are clearly described and consider possible confounds, especially factors that might contribute to signal attenuation.*

*1) Along these lines, we want the revision to specify the input impedance of the amplifiers and the typical electrode impedance.*

The following sentence was added to the Methods section: “The input impedance of the amplifier is above 100 MOhm, and electrode impedance was kept below 75 kOhm during the recordings. Variation in impedance of up to 50 kOhm across electrodes would contribute at most 0.05 percent error in the voltage measurements.”

*It is stated that electrodes were considered high impedance if there was strong 60 Hz noise on these channels, but it is unclear if this approach eliminated electrodes with low enough impedance to prevent 60 Hz noise, but high enough impedance to cause an attenuation of the recorded voltages. Obviously, any unintended voltage attenuation, while not critical for intracranial EEG, can affect all estimates of E-field strength and tissue conductivities in this study.*

We further explain: “Voltage variations due to varying impedance across electrodes are therefore negligible given other sources of variation, predominantly movement leading to altered current paths and transient noise.”

*Furthermore, it is implied that there was clipping of the signal in some electrodes (data from them were not included in the analysis). During clipping did the input impedance of the recording amplifier decrease and could this produce current stimulus flow though the recording electrodes?*

Had this happened, we would have seen obvious artefacts in other electrodes that were not clipped. There were no such collateral artefacts in unclipped electrodes. Thus, clipping did not affect our voltage estimates in other electrodes.

*2) There is a preprint available on bioaRxiv that involves similar experiments on two patients and two NHP (http://biorxiv.org/content/early/2016/05/18/053892, Opitz et al.). The results reported in that preliminary report could be compared to the present work. Given the great interest we anticipate in your work and this other study, it would be useful to comment on the similarities and differences between the experimental results in two studies. Two issues that come up when comparing the studies are 1) possible frequency-dependent effects and the manner in which these were analyzed, and 2) differences in how electric fields are estimated. For example, for electrodes in the interior of the grid why not use all neighbors instead of just the closest one?*

We added the following paragraph in the Discussion: “A recent study has also reported the electric field produced by TES as measured by stereotactic EEG electrodes in two epilepsy surgical patients (Opitz et al., 2016). […] In our equipment we find a 25% drop which we attribute to a non-uniform gain of the equipment (in saline we measured a gain difference of 25% between 1 Hz and 100 Hz).”

On the second topic raised here, how electric fields are estimated, we note that Opitz and us both estimate the projected electric fields at the electrodes by taking local differences. There are not substantial differences in the methods. They use a three point method, we use a two point method. Three point has the advantage that one combines information from one more point giving possibly less measurement error. On the other hand, fields are not uniform, thus, by combining more distant electrodes one makes potentially a larger approximation error. For that reason, we opted to use only the closest electrode. We added the following to the Methods: “Since electric fields are non-uniform, only local estimates of projected fields were measured by taking the difference only between the closest electrodes. Other field directions and averages could be obtained by looking at more distant pairs, at the cost of making systematic errors due to non-uniformity.”

*II) Concerns with the modeling work: We see the modeling work as a nice complement to the experimental data. We recognize that this work entails complex models, involving a number of assumptions. Below we summarize major issues to consider in revising this section. Most involve clarifying the methods and potential confounds of the models, and carefully qualifying the conclusions based on these models. In some cases, we would recommend conducting additional simulations and/or analysis since this will reduce qualifications you would have to place on the conclusions.*

Based on these comments, we did significant additional modeling work to address factors that may have affected model accuracy:

1) We more carefully placed the stimulating electrodes based on photographs of the montages during experimentation, where available.

2) Following the reviewers’ suggestion we extended the model to include lower head and neck.

3) We more carefully segmented the eye-sockets to prevent gaps in skull bones resulting from the automated segmentation of very thin structures.

These changes improve the modeling results considerably from 0.76 to 0.89 (correlation of measured and predicted electric fields). We then re-ran the optimization to obtain calibrated models now tuning skull, scalp, and brain conductivity by minimizing the difference between measured and predicted electric field. However, following your valid concern about bias, we no longer used the “optimal” conductivities in the model comparison. We make all these points clear in the new manuscript. As a result of these changes Figure 1,Figure 2,Figure 5–Figure 9 have been updated with the new results. The Abstract also reflects this new information: “When individual whole-head anatomy is considered, the predicted electric field magnitudes correlate with the recorded values in cortical (r=0.89) and depth (r=0.84) electrodes.” Also in the Discussion: “However, extending the FOV to include the entire head and neck significantly improves prediction accuracy.” We discuss these items in more detail for each comment below.

*3) The study concludes that modeling anatomical detail such as the skull layers and the white matter anisotropy does not improve the model predictions. It seems, however, that the models were optimized only for the "realistic model" without the respective anatomical refinements. This should be clarified, and if this is indeed the case, the comparisons are biased since one side of the comparison is optimized and the other is not.*

Thank you for pointing out this bias in the comparison. We now do this comparison using fixed conductivity values used in the literature, i.e., no optimization in either the “realistic models” or the models with skull layers or white matter anisotropy. We focused in the comparison on the accuracy of electric field distributions, which is not as sensitive to the specific choice of conductivity values. Note also that we cannot perform a statistical evaluation for the DTI data as we only have 3 subjects with DTI.

*Furthermore, the white matter conductivity anisotropy affects mostly the E-field in the white matter, while most of the voltage recordings were not in the white matter, which could affect efforts to optimize the anisotropic model fit to the data.*

We did not optimize the anisotropic model to fit the data. Solving the anisotropic model is time-consuming (1 week per evaluation), so optimizing for conductivity values is prohibitive as it would require evaluating the model hundreds of times, resulting in years of computing time. Regardless, optimization has been removed from the model comparison aspect, so this bias should no longer be there. We add the following to the Discussion on the issue of DTI: “We could not evaluate the benefits of DTI modeling statistically, as we had only 3 subjects with DTI data. But for these 3 subjects there was no obvious benefit from adding DTI. […] The various approaches proposed in the literature have been evaluated using the recorded data here, and there does not seem to be any difference between them.”

*4) (also relevant to experimental results) The selected montage (e.g., inferior position), combined with the large inter-electrode distance increases the significance of conduction paths in the lower half of the head. These include regions with complex anatomy and conductivity profiles such as the eyes, orbits, optic nerve, neck, nasal cavity, etc., as well as the truncation of the head and neck (as seen in Figure 4).*

You make an important point here. We are aware that the lower head may significantly affect the results. Unfortunately this data was not available as the field of view is routinely restricted in clinical MRI scans. Nonetheless, we have now made an effort to address this by adding the lower part of the head using a standard head model (having to do all the optimization for these new models over again was the main delay for the resubmission). Indeed, we found that the predictions improved when extending the field of view. We have added the following text to the Results: “Many have argued that it is important to incorporate the entire head anatomy down to the neck. […] By extending FOV with a standard head to capture the anatomy down to the neck we demonstrated that capturing current flows through the entire head significantly improved the modeling results.”

*Furthermore, only a limited set of tissues are modeled, and other potentially significant ones are omitted such as muscle, fat, and sub-compartments of the eye. Discussion should be provided concerning the extent these regions affect the accuracy of the electric field simulation as well as the derived "optimal" conductivity of the scalp and skull.*

Agreed. Unfortunately there is no limit to the level of detail one could incorporate in a model. given our limited capacity to do absolutely all that is possible, we have opted to make the data publicly available. We add the following discussion giving reference to what has been done in the literature on these patients. “Wagner et al. (2014) argues that modeling the skull as multiple compartments only has a meaningful effect when there is significant volume of cancelous bone along the current path. […] To allow testing of such refinements we have made the MRIs and recorded voltage values freely available to the public (at http://dx.doi.org/10.6080/K0XW4GQ1).”

*5) The derived "optimal" conductivity values for scalp (mean = 3.61 V/m) are very high, being an order of magnitude above values in the literature and more than twice the measured conductivity of CSF and physiological saline (both ~ 1.6 S/m). It is unclear what physical mechanism could possibly confer such high conductivity. While this is acknowledged by the authors, this is an issue of concern. Since they are based on a fit of a complicated model with just two free parameters to a limited data set, it is entirely possible that the optimized conductivity values result from unrecognized limitations of the model and/or recordings. For example, truncation of the head and neck (as seen in Figure 4) or other anatomical inaccuracies in the lower portion of the head could artificially strengthen the modelled current flow in the top portion of the head, requiring higher scalp conductivity values to match the recorded potentials.*

We fully agree on all counts here. We now acknowledge in the Discussion that the “optimal” conductivity values should be regarded with care (more on that later). We agree with the concerns and so we repeated optimization now on the heads with extended field of view. The new values indeed seem to be more meaningful. We now write in the Results section: “We calibrated the models by adjusting conductivity values for each individual model with the goal of minimizing the mean square error between predicted and measured field projections (see 'Conductivity optimization'). […] Note that the conductivities of tissues are optimized jointly, i.e., they are adjusted simultaneously per evaluation of the cost function.”

*Further, the optimized conductivity values may depend on the location of the tES electrodes. Adopting the "optimal" values from this study in general for tES (and potentially other) simulations may produce errors when the model context changes. We would like you to consider quantifying the impact of these model limitations; for example, test the impact of head truncation and reduction of the number of segmented tissue compartments in one of the full head and neck models. In addition, the revision should be explicit about these limitations and impact on inferences to be drawn from this work.*

We fully agree. We have performed the evaluation you requested here, removed any claims from the Discussion that encourage generalization of these results beyond the current type of models, and instead have added the following qualifiers in the Discussion: “Generally, the model fitting appears to be under-constrained as differing parameters gave similar goodness of fit (Figure 8). […] Thus, “optimal” values differ considerably across subjects. Nonetheless, the median values we report here appear to be a good compromise for predicting field distributions when individual calibration is not possible.”

In the Results we write: “Note that all model comparisons were performed using the same fixed conductivity values (from the literature) to avoid biasing the comparisons with optimized parameters. Additionally, model comparisons focused on the accuracy of the spatial distributions, as captured by the correlation values, as they are not strongly impacted by the specific choice of conductivities (Figure 8).”

In the Methods we also write: “Note that when comparing across different categories of models listed above, literature conductivity values (see 'Conductivity optimization') were used when solving the models. This is to avoid any potential bias that may be introduced if using the optimized conductivities by fitting the model to the recorded data ('Conductivity optimization').”

*6) The paper attempts to solve two mathematical problems: (A) An inverse problem in which they the estimate conductivity (scalp and skull only) given the voltage measurements noted above and (B) A forward problem in which they calculate the scalar potential field and the electric field everywhere (or at least potentially so) within the head given the previously estimated conductivity and assumptions of their model (6 regions delineated by anatomical MRI with each region being of spatially constant conductivity). The solution of problem (A) is attempted through a minimization of error between measured and predicted values as given by Equation 1. The solution of problem (B) is obtained through the Abaqus software. For the estimates of the electric field to be meaningful, a reasonable accuracy estimate must be associated with the estimate. No such estimate is given. To do so would require at least the following: (i) A reasonable estimate of the error produced in results of problems (A) and (B) that is associated with assumption that the conductivity is spatially constant throughout each of the 6 segmented regions and (ii) A reasonable estimate of the error in the results of problems (A) and (B) associated the measurement and segmentation of the six regions. In addition, known instrumental error must be propagated through both problems as well.*

The sources of errors are many, some of which could be potentially quantified. For example, the uncertainty in the empirical measurements of conductivities. However, even if error bars were available (they are not in the literature, but one could stipulate a best guess) from the preceding discussion on conductivities, it is clear that any one conductivity aggregates in a single number an entire set of simplifications, so we are not sure how meaningful an error propagation would be. Other sources of errors are the estimates of electrode locations. Of course we could do an exhaustive sensitivity analysis on the model for all input parameters to the model (conductivity, locations, etc.). But we feel that ultimately the comparison with the measurements provides a good sense of the overall accuracy of the entire modeling process (B). At the end of the day we provide a deterministic model prediction and evaluate it against measurement. The performance metrics we report give a measure of success of this modeling enterprise. Now as to problem (A). We have now added a number of caveats to this “inverse problem” above, essentially confirming that we are not really inverting, but simply fitting parameters. Additionally, process (B) is independent of problem (A) as we now do all model comparisons with a fixed set of conductivities. Thus, we do see the need to “propagate” errors from (B) to (A), as we have now effectively separated the two.

*7) Why aren't the correlation values between measured and predicted fields better than are reported here? If the solution to problem (A) is unique and if the voltage measurements are consistent with the quasi-static field equations, then why don't the results of problem (B) reproduce the measurements of voltages (albeit with some presumably small measurement error discrepancies)?*

We discuss this now as follows: “The overall correlation of predicted and measured fields is 0.89 for cortical electrodes, and 0.84 for depth electrodes (Figure 5). […] Finally, openings in the eye socket such as optical foramen and orbital fissure are hard to establish and could serve to guide significant current for the present electrode montages.”

*III) Concerns with the organization and clarity of the paper:*

*8) In general, there is concern with organization, clarity, precision, and to some extent discussion. The current organization is hard to follow. There are methods descriptions mixed into the Results section, while at the same time parameters reported on in Results (e.g. "s", "r", "t(n)" – the latter does not even seem to be explained in Methods)) without any description at that point of what they mean or any mention that they are described in Methods. Please review your manuscript from the reader point of view, rigorously stripping methods material from Results and inserting descriptive phrases and referrals to Methods in Results where needed. The statistics should be described more carefully in the Methods section.*

Without listing all edits in detail, we can assure the reviewer and editor, that we have made in the revised manuscript a significant effort to edit the paper for clarity and completeness. We assured that the Result is clear and self-contained, and that the Methods are clear and complete.

*A number of experimental details are missing or not clear, including the specific criteria by which some patients were excluded and also by which electrodes were excluded within a given patient (since the latter is a very significant fraction of the total number of electrodes this seems particularly important) and what the range of electrodes was across all subjects (we are only given the total).*

Some patients were excluded: we edited the corresponding text in the Methods: “We enrolled 12 patients between December 2013 to June 2016 (P03--P014). […] Therefore, we have included 10 subjects in this paper (P03--P011, P014).”

Electrodes were excluded within a given patient: we added this sentence: “In total we recorded from 1380 intracranial electrodes: P03 (124), P04 (124), P05 (124), P06 (124), P07 (80), P08 (120), P09 (250), P010 (124), P011 (118), P014 (192).” Details on why some electrodes were excluded are described in section 'Voltage and projected electric field measurements', we also added this sentence there: “Those electrodes with strong 60 Hz interference and evident clipping (strong harmonics in the harmonic fit) were discarded and were not used for model validation. In total we were able to use 1205 electrodes among a total of 1380 electrodes: P03 (118/124), P04 (112/124), P05 (118/124), P06 (117/124), P07 (78/80), P08 (116/120), P09 (184/250), P010 (101/124), P011 (111/118), P014 (150/192).”

*Make sure the manuscript clearly describes when and why particular sections/graphs exclude some of the participants.*

We make now clear in all instances which patients could be used for model comparison and optimization. P04 and P06 could not be used for determining the benefits of extending FOV as we were not able to extend it for practical limitations (see above). All patients are now optimized in conductivity, and DTI could only be performed on 3 patients for which we had their data. We carefully reviewed the manuscript to be sure these numbers and limitations are clearly stated.

*There are a number of places were the grammar needs work, including instances where there is disagreement of subject and verb, missing prepositions, missing definite or indefinite articles, etc. Please review the manuscript to be as precise as possible (e.g., what does "as important as commonly thought" in the subsection “Relative merit of various model refinements”?).*

We had a native speaker carefully edit the grammar.

[Editors' note: further revisions were requested prior to acceptance, as described below.]

*There remains one important issue to be addressed. While you have provided E field predictions, the manuscript does not provide an associated estimate of the error term for these predictions. A complete response would address the following:*

*1) For conductivities that are allowed to vary in the current fitting procedure: Provide the range of conductivity values that work nearly as well as those used in the current version.*

To provide a sense of the sensitivity of the modeling performance on these parameters we performed further simulations. Specifically, we estimated the variance of the best-fit values by looking at conductivities that are close to the optimum and estimating the Cramer-Rao bound. We did this for every conductivity value and report the results for each subject in a new Figure 8—figure supplement 1.

*2) For conductivities that have been fixed in the fitting procedure: Provide a range of conductivity values that fall within reasonable estimates of error for those values.*

Here estimation variance cannot be computed as above (as the Cramer-Rao bound only applies at the optimum). Instead we varied the literature values by +/- 10% and evaluated changes in model performance. These results are also reported in the new Figure 8—figure supplement 1.

*3) A range of compartment segmentations taking into account a reasonable estimate of error for the accuracy of the segmentation methods.*

We acknowledge this as a general limitation of the modeling process. Since it is hard to perform a sensitivity analysis based on segmentation variations, we added a discussion based on the extensive modeling results that are already in the manuscript. Specifically, we think that Figure 9 provides insights on this question as it compares a number of model variations. We now write in the Discussion: “A general sense for the sensitivity of the model accuracy to errors in segmentation can be obtained by comparing the prediction performance of the intact models and the realistic models (IM vs. RM in Figure 9). Despite multiple refinements to the segmentation in RM, there was no consistent performance improvement across subjects.”

For details on the new results and text, please refer to the new manuscript.

We hope that these steps adequately address your legitimate questions, and point out once more that we provide extensive caveats in the manuscript on the interpretation of these best-fit conductivity values.

[Editors' note: further revisions were requested prior to acceptance, as described below.]

*Just two revision requests to consider. I am confident that you will be able to address them quickly and then we can move to a final decision after the Reviewing Editor has a chance to take a look at the revision.*

*1) Reviewer 1 has two suggestions for improving* Figure 5—figure supplement 1*:a) In Figure 8—figure supplement 1, logarithmic spacing of the y axis could be considered given the wide range of the values (over a decade) and the seeming proportional behavior of the C-R bound error bars.b) In Figure 8—figure supplement 1, add lines connecting the two points for each subject in the corresponding color to allow easier visual inspection of the local sensitivity.*

We have modified the supplementary figure as requested and added the following paragraph to the Discussion section summarizing the findings on estimation of error:

“To provide a sense of how sensitive model predictions are to choices of the conductivity value we computed the estimation accuracy for the best-fit values. […] Thus, it may not be surprising that optimal values differ across subjects by as much as one order of magnitude, yet a single median value provides reasonable accuracy in terms of spatial distributions across all subjects (Figure 8).”